behaviour/cognition/psychology

non-verbal reasoning, adolescence, speed–accuracy trade-off, matrix reasoning

**Author for correspondence:**
Gabriele Chierchia
e-mail: g.chierchia@ucl.ac.uk

†These authors are joint first authors.

# The matrix reasoning item bank (MaRs-IB): novel, open-access abstract reasoning items for adolescents and adults

Gabriele Chierchia[1,†], Delia Fuhrmann[1,2,†], Lisa J. Knoll[1], Blanca Piera Pi-Sunyer[1], Ashok L. Sakhardande[1,3] and Sarah-Jayne Blakemore[1,4]

[1]Institute of Cognitive Neuroscience, University College London, London, UK
[2]MRC Cognition and Brain Science Unit, University of Cambridge, Cambridge, UK
[3]Centre for Community Child Health, Murdoch Children's Research Institute, Melbourne, Australia
[4]Department of Psychology, University of Cambridge, Cambridge, UK

(iD) GC, 0000-0002-5623-4573; DF, 0000-0003-4678-8828;
BPP-S, 0000-0002-6707-9943; S-JB, 0000-0002-1690-2805

Existing non-verbal ability tests are typically protected by copyright, preventing them from being freely adapted or computerized. Working towards an open science framework, we provide 80 novel, open-access abstract reasoning items, an online implementation and item-level data from 659 participants aged between 11 and 33 years: the matrix reasoning item bank (MaRs-IB). Each MaRs-IB item consists of an incomplete matrix containing abstract shapes. Participants complete the matrices by identifying relationships between the shapes. Our data demonstrate age differences in non-verbal reasoning accuracy, which increased during adolescence and stabilized in early adulthood. There was a slight linear increase in response times with age, resulting in a peak in efficiency (i.e. a measure combining speed and accuracy) in late adolescence. Overall, the data suggest that the MaRs-IB is sensitive to developmental differences in reasoning accuracy. Further psychometric validation is recommended.

## 1. Introduction

Abstract reasoning is the ability to solve novel problems without task-specific knowledge and a core mechanism of human learning [1]. Abstract reasoning is closely related to more fundamental cognitive functions such as processing speed [2] and working memory [3]. It is also predictive of a number of mental health symptoms [4] and educational attainment [5].

Due to this predictive power and because it is typically neither trained, nor taught, abstract reasoning is often assessed as part of fluid intelligence tests, including the Wechsler Abbreviated Scale of Intelligence [6], Cattell Culture Faire Intelligence Test [7] and Raven's Progressive Matrices [8]. These IQ tests provide reliable, and well-validated abstract reasoning items [9] and are extensively used in clinical, educational, occupational and research settings [10,11]. However, these tasks are not free-to-use, and copyright usually prevents these pen-and-paper tasks from being adapted into computerized tasks. Notable exceptions here are the freely available web-based 20-item Hagen Matrices Test [12] and the 11 matrix reasoning items of the public-domain International Cognitive Ability Resource [13]. These tasks include a very limited number of items, however, and, to our knowledge, are not currently validated in developmental populations.

This limits the usefulness of existing tasks for research studies requiring large sets of computerized items, such as online, neuroimaging and longitudinal studies. In addition, the largely analogue task administration means that existing normative data for these tasks are typically limited to accuracy data and do not include response times. The latter is an important practical and theoretical limitation because of the well-known relationship between abstract reasoning and processing speed [2,14,15] and because of the potential existence of speed–accuracy trade-offs [16].

Speed–accuracy trade-offs were first demonstrated experimentally over a century ago by Woodworth [17] and Martin & Müller [18] for simple motor tasks and are ubiquitous in mental computations of all kinds and across species [16]. According to sequential sampling theories, speed–accuracy trade-offs can be viewed not as experimental noise but rather as a direct result of sequential sampling of evidence [19,20]. The more time spent on a mental computation, the more information can be accumulated. The point at which a decision is made therefore contains rich information about an individual's internal goals, as well as external task demands [16,19]. This is particularly pertinent in developmental studies.

Abstract reasoning capacity increases during childhood and adolescence and peaks in early adulthood [21–23]. These changes in reasoning performance during development have been linked to the protracted maturation of the frontal cortex [21,24,25]. During the same developmental period, impulsivity decreases markedly [26,27], raising the possibility that increases in reasoning scores could reflect either true increases in reasoning capacity, a decrease in impulsivity or both. In other words, there may be a developmental speed–accuracy trade-off.

To address these gaps in the literature, here we provide a set of 80 abstract reasoning items, each one in three shape variants, as well as accuracy and response time item-level data from a large sample of 659 participants aged 11–33 years. These items are freely available for non-commercial purposes (https://osf.io/g96f4/) and can be selected by researchers to yield age-appropriate tasks of custom difficulty and duration.

The design of the matrix reasoning item bank (MaRs-IB) was similar to Raven's matrices [8]. The items consisted of a three-by-three matrix containing abstract shapes in eight out of nine cells. Participants deduced relationships across the eight shapes, which could vary across four relations: colour, size, position and shape. Participants then identified the missing shape from an array of four options. See https://gorilla.sc/openmaterials/36164 for a demonstration of the items. The dimensionality of each item is linked to the number of relations changing in the matrix. While one-relational changes (e.g. a colour change, figure 1a) are typically easy to identify, higher-order relational changes become increasingly difficult (e.g. a three-relational change of shape, colour and position, figure 1b).

The items were originally developed as part of an online training study [28]. Here, we analyse data collected before training using adolescents and adults (N = 659, aged 11–33 years) to provide accuracy and response time data at the sample level, age group level and item level. We also assessed age differences in accuracy, response times, efficiency and productivity in the MaRs-IB to provide insights into developmental differences in response patterns as well as potential speed–accuracy trade-offs. Our aim here is to introduce a novel, open-access item bank of abstract reasoning items for studies that include adolescents and adults, and to provide a preliminary investigation of their psychometric characteristics.

# 2. Methods

## 2.1. Participants

Here, we analysed data from baseline assessments of a larger training study [28]. The research team recruited 821 participants from 16 schools in the London area and through UCL participant pools and posters around campus. Of this sample, 659 participants were included in the current analysis (396 females and 263 males, mean age at baseline = 16.21, s.e. = 0.16, age range = 11.27–33.15 years). The

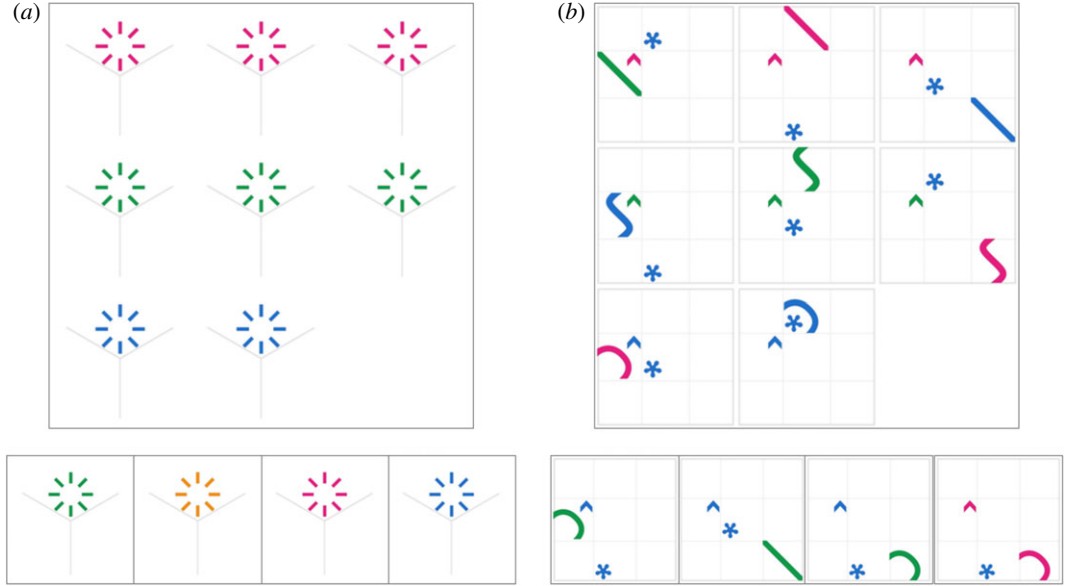

**Figure 1.** Example items from the MaRs-IB. (*a*) A simple item containing a one-relational change (i.e. only the colour changes) and answer options. The fourth option is the correct solution. (*b*) A harder item containing a three-relational change (i.e. shape, colour and position change). The third option is the correct solution.

**Table 1.** Demographics of the sample.

| age group | sample size |
|---|---|
| younger adolescents (aged 11.27–13.39) | total: 185 |
| | females: 118 |
| | males: 67 |
| mid-adolescents (aged 13.40–15.91) | total: 184 |
| | females: 89 |
| | males: 95 |
| older adolescents (aged 15.93–17.99) | total: 184 |
| | females: 108 |
| | males: 76 |
| adults (aged 18.00–33.15) | total: 106 |
| | females: 81 |
| | males: 25 |

exclusion criteria were: missing relational reasoning data at baseline ($N = 5$); missing parental consent for adolescents ($N = 123$) and developmental conditions such as attention-deficit hyperactivity disorder and dyslexia ($N = 34$). For the training study, adolescents were split into three age groups of equal size and adults were included as a fourth age group (table 1). We report data for each of these age groups to provide researchers with guidelines for different age groups. We also replicated our analyses using age as a continuous variable.

A follow-up study was conducted to assess the convergent validity of the MaRs-IB. A power analysis suggested that 38 participants are sufficient to detect a correlation of 0.5 at 90% power. For this study, we thus initially recruited 50 participants, with a further 50 participants tested upon reviewer request (total $N = 100$, 73 females, 27 males, mean age = 23.95, s.e. = 0.35, age range 19–35). The results of the initial analysis ($N = 50$) are available in the electronic supplementary material for transparency (see supplementary results). Studies were carried out in accordance with UCL Research Ethics Guidelines and approved by the UCL Research Ethics Committee. Informed assent or informed consent was obtained from all participants included in this study. The studies were not formally preregistered. All

materials and item-level data have been made available on a permanent third-party archive (https://osf.io/g96f4/); requests for the item-level data can be sent via email to sjblakemore@psychol.cam.ac.uk. The data analysed here were derived from a training study, which was powered to detect small-to-medium effect sizes [28].

## 2.2. Testing procedure

The testing procedure was previously described by Knoll *et al*. [28]. Participants were tested on a battery of tasks including one containing MaRs-IB items. Testing was delivered via an online platform developed by the research team and a software company (www.cauldron.sc).

Adolescent participants completed the testing session in groups of 3–48 in school, while adult participants were tested in groups of 1–15 in a university computer room. Participants used computers or tablets. Responses were recorded using a mouse or touchpad. The order of the tasks was counterbalanced between testing groups.

An experimenter gave instructions before the task. Participants then completed practice items until three were completed correctly. More than three practice items were required by 121 participants. On average, these 121 participants needed 4.46 (s.e. = 0.10) practice items to proceed to the task and never more than 10. All participants completed three practice items successfully during the testing session and proceeded to the task. In all items (i.e. both practice and test items), participants were given visual feedback on their performance.

## 2.3. MaRs-IB design

Each MaRs-IB item consisted of a $3 \times 3$ matrix. Eight of the nine resulting cells contained an abstract shape, while one cell on the bottom right-hand side of the matrix was empty. The participants' task was to complete the matrix by finding the missing shape among four possible alternatives (see figure 1 for examples). To select the correct missing shape, participants had to deduce relationships between the shapes of the matrix. These shape characteristics varied along four dimensions: shape, colour, size and position in the matrix.

Items began with a 500 ms fixation cross, followed by a 100 ms white screen. Participants were then given up to 30 s to complete an item. After 25 s a clock appeared, indicating that 5 s remained before the next item began. An item ended when participants responded, or after 30 s had elapsed without response. For each participant, puzzles were taken from one of three test forms of 80 items. These test forms were created to prevent familiarity effects in future testing sessions of a training study and they were generated by drawing from one of three shape sets, in a counterbalanced fashion. The items in each test form thus tested the same abstract reasoning problems and differed only in the exact shapes used (see electronic supplementary material, figure S1 for examples). These shape set variants are also available in the online repository (https://osf.io/g96f4/). Items were presented in the same order for all participants, starting with five simple items intended to familiarize participants with the task. Participants completed the items for 8 minutes but were not informed of the total number of items and were not required, or expected, to complete 80 items in 8 min. The only time constraint stated was to provide a response to each item within 30 seconds. If a participant completed 80 items within 8 min, the items were presented again in the same order, but responses were not analysed.

Two different algorithmic strategies were employed to generate the target and distractor items in the answer options: a *minimal* difference strategy and a *paired* difference strategy. These strategies were used to control for the possibility that participants may solve the puzzle in unintended ways. Specifically, the distractor strategies were used to counterbalance the risk of pop-out effects (where the target 'pops out' and can be identified by reasoning in fewer dimensions than intended) with the risk that participants could reason from the answer options alone, without looking at the puzzle itself. Note that it is not possible to prevent both possibilities at the same time for three- or higher dimensional puzzles when using only four answer options. The minimal difference algorithm created distractors as variations of the target (see electronic supplementary material, figure S2). This prevented pop-out effects but theoretically allowed participants to solve the puzzle by looking at the answer options only. A paired difference algorithm created distractors that had at least one component in common with the target (electronic supplementary material, figure S2). This could theoretically induce pop-out effects but prevented participants from reasoning from the answer options alone. To counterbalance these risks, each item was pseudo-randomly assigned one of these two distractor strategies and we tested whether the distractor strategies affected MaRs-IB performance in our analyses (see below).

## 2.4. Convergent validity study

In the follow-up study, we asked participants to complete the MaRs-IB, as well as an existing task: the 'International Cognitive Ability Resource' (ICAR) [29]. The latter includes four main tasks: matrix reasoning, series completion, spatial rotations and verbal reasoning. The ICAR is relatively novel. While more traditional tasks are more thoroughly validated, the ICAR is computer-based and has been tested on a large sample of individuals (more than 90 000). The ICAR has also been indirectly related (via the Shipley-2) to more traditional tests such as the Wechsler Adult Intelligence Scale. We particularly focused on the matrix reasoning items of the ICAR, as these most closely resemble the MaRs-IB items and other tests like Raven's progressive matrices. The MaRs-IB and ICAR were administered in counterbalanced order.

## 2.5. Mars-IB items, data and analyses

All 80 MaRs-IB items in their test forms are freely available for non-commercial use and can be downloaded from https://osf.io/g96f4/. The repository also contains item-level dimensionality, accuracy and response time data for each age group, test form and shape set, which can be used by researchers to adapt the items to tasks of different duration and difficulty. Two further, colour vision deficiency-friendly sets of stimuli, can also be obtained from the repository.

Here, we provide sample- and age group-level accuracy and median response time data on correct items. We also analysed the effect of age on accuracy, response times, productivity and inverse efficiency. Productivity was operationally defined as the number of items completed. Inverse efficiency was calculated as median response times divided by accuracy [30].

Items with a response time under 250 ms were excluded from all analyses. We modelled each of these four dependent variables using mixed models because these allowed to accurately partition the error terms according to the hierarchical structure of the data (e.g. item-level data were clustered by participants, which were in turn nested in schools) [31]. We used generalized linear mixed models (GLMMs) for accuracy and response times, and linear mixed models (LMMs) for inverse efficiency using lme4 [31] in R [32]. GLMMs were used to model raw item-level data, with participant ID and school or university as nested random intercepts and item code as crossed random intercept. Accuracy and response times were modelled at the item level after the removal of incomplete items. Accuracy data were modelled using the binomial distribution with a logit link function while response times were modelled using the Gamma distribution with a log link function [33]. Productivity (the number of items completed) was necessarily an aggregate measure. To calculate inverse efficiency, data also had to be aggregated across items for each participant. LMMs estimating the number of items completed and inverse efficiency therefore included only school or university as a random intercept. All models included fixed effects for either age group as a categorical Helmert-coded variable or orthogonal polynomials of age with the linear, quadratic and cubic components. In exploratory models, we additionally included Helmert-coded gender and the interaction between age group and gender as fixed effects.

To assess the equivalence of the forms of the MaRs-IB, we used exploratory GLMMs specified as described above to investigate the effect of test form (test form 1, 2 or 3) on accuracy and response times. Moreover, given that test forms were created by pseudo-randomly drawing items from one of three different shape sets, and solutions were pseudo-randomly generated according to one of two distractor strategies, we additionally investigated whether such test form constituents might also independently affect accuracy and response times. This information may be useful for researchers interested in shaping novel test forms (e.g. by differently combining shape sets and distractor strategies). As a secondary analysis of equivalency, we thus assessed whether distractor type (minimal or paired difference) and shape set (shape set 1, 2 or 3) might also affect accuracy and response times.

To assess response processes, we investigated how accuracy and response times were affected by item dimensionality (a score ranging between 1 and 8, reflecting the number of dimensions changing in a puzzle), using GLMMs as specified above. For the analysis of dimensionality only, the item-related random intercept was removed because of multicollinearity between each item and the associated dimensionality. Main effects and interactions were inspected using omnibus Type III Wald $\chi^2$ tests. Planned and *post hoc* comparisons were performed using the emmeans package [34] and Bonferroni corrected for multiple comparisons.

The analyses above were based on completed items. This excludes (i) trials in which participants timed out and (ii) trials that were not reached. Our data stem from a previous study that aimed to address a different research question. The data are therefore not optimized for psychometric validation. In

particular, participants completed varying number of items, and item-level accuracy data are based on a different number of responses for each item, with completion rates dropping for later items. This may limit their comparability. To address this issue, we used mixed models (as described above), which are able to deal with unbalanced data by modelling performance at the trial level, rather than participant level. Furthermore, to provide preliminary insight into more traditional psychometric approaches to item functioning (i.e. internal consistency, biserial correlations, $p$-values and differential item functioning (DIF)), we focused on a subset of data that involved no variability in completion rate.

To investigate possible DIF, we used logistic regression [35], as implemented in the 'difR' package in R [36]. We tested for both uniform and non-uniform DIF, correcting for multiple comparisons using the Benjamini–Hochberg method as recommended by Kim & Oshima [37]. As potential sources of DIF, we focused on age (used as both categorical and continuous) and gender.

## 2.6. Convergent validity analysis

To investigate convergent validity in the follow-up study, we inspected the product-moment correlation between performance in the MaRs-IB and in the matrix reasoning items of the ICAR. Although the assumptions for correcting for range restrictions could not be fully assessed, our sample was tested in a highly selective university. We therefore followed Condon & Revelle [29] and assessed the possible presence of range restriction by comparing the standard deviations of the matrix reasoning items in our sample and theirs. We obtained the latter by combining the standard deviations for the ICAR matrix reasoning items (listed in table 2 of p. 55 of the study by Condon & Revelle [29]). Range correction was performed with the 'rangeCorrection' function of the Psych package in R [38].

# 3. Results

## 3.1. Overall performance on the MaRs-IB and preliminary reliability measures

The mean accuracy was 69.15% (s.e. = 0.65%, range = 24.24–100.00%), and the median response time on correct items was 7914 ms (interquartile range (IQR) = 4063 ms). On average, participants completed 32.78 unique items (s.e. = 0.41, range = 13–80) of the MaRs-IB within 8 min. Timing out occurred in only 2% of items. To assess the presence of potential ceiling and floor effects, we inspected skew and kurtosis of response accuracy as well as the rate of chance performance and perfect performance across participants. A skew of −0.41 and a kurtosis of 2.39 suggested that participants' scores fell reasonably close to the normal distribution. Two participants performed below chance (i.e. ≤25% correct items), while eight participants responded correctly on all items, suggesting that 98.48% of participants were above minimum and below maximum accuracy. These results suggest that there were no obvious floor or ceiling effects.

To assess internal consistency, biserial correlations, $p$-values and DIF, we focused on a subset of items for which there was no variability in completion rate. Specifically, we focused on the 25 items following item 5 (since the first five items were deliberately easier and involved virtually no variance in performance; see MaRs-IB design). For this analysis, we also included trials in which participants had timed out (labelling them as incorrect), as well as those in which response latencies were shorter than 250 ms (only two of which were correct). This resulted in a subset of 25 items that were completed by $N = 349$ participants (at least $N = 70$ per age group). To assess internal consistency on this subset of items, we computed Kuder–Richardson 20 formula, which resulted in a KR-20 of 0.78 (0.95% confidence interval (CI) [0.78, 0.79]). The mean item-total biserial correlation for the same items was 0.32 (s.e. = 0.02) (biserial correlation of each item is available in electronic supplementary material, table S2). The mean item $p$-value of this subset of items (i.e. the proportion of participants that answered correctly to each item) was 0.59 (s.e. = 0.04) ($p$-values of each item are available in the electronic supplementary material, tables S1 and S2). Note that this mean $p$-value differs slightly from the mean accuracy value reported above because it refers to the subset of items, not to the full dataset. None of these items displayed uniform or non-uniform DIF, suggesting they are unbiased relative to the age groups studied here and relative to gender. Researchers may want to preferentially use this subset of items for which a more in-depth psychometric analysis is available.

To assess the reliability of the items, we inspected its split-half, test–retest reliability in the full dataset. The Spearman–Brown split-half reliability coefficient was computed on the raw item-level data and was 0.82 overall (younger adolescents: 0.82, mid-adolescents: 0.81, older adolescents: 0.71, adults: 0.80). To

assess test–retest reliability, we focused on a subset of participants ($N = 218$) who were assessed on the MaRs-IB twice, 35 days apart on average (s.e. = 1 day, range = 21–52 days). During this time they trained in an unrelated face-perception task (see [28] for details). Pearson's product–moment correlation in accuracy between the two time points was $r = 0.71$ ($t(183) = 13.58$, $p < 0.001$), suggesting acceptable test–retest reliability. The test–retest relationship did not interact with age ($\chi^2(3) = 2.20$, $p = 0.531$).

To assess the equivalency of the MaRs-IB forms, we inspected accuracy and response times across different test forms, distractor types and shape sets. The item information of one participant (but not their performance-related information) was lost due to a technical error. Test forms or distractor types did not affect accuracy or response times (test form accuracy: $\chi^2(2) = 0.62$, $p = 0.733$; test form response times: $\chi^2(2) = 4.85$, $p = 0.088$; distractor type accuracy: $\chi^2(1) = 0.88$, $p = 0.347$; distractor type response times: $\chi^2(1) = 0.48$, $p = 0.491$), suggesting good equivalency of the test forms and of the distractor strategies. Shape sets had no significant impact on accuracy ($\chi^2(2) = 4.85$, $p = 0.088$), but there was a significant effect on response times ($\chi^2(2) = 12.25$, $p = 0.002$). *Post hoc* comparisons suggested that participants showed reduced response times for shape set 1 compared to shape set 2 ($z = -3.46$, $p_{Bonf.} = 0.002$), while there were no differences between shape sets 1 and 3 ($z = -2.16$, $p_{Bonf.} = 0.091$) or shape sets 2 and 3 ($z = 1.28$, $p_{Bonf.} = 0.601$). These response time differences were unexpected. We speculate that they may be due to low-level perceptual features of the different shape sets. See electronic supplementary material, tables S3–S5 for descriptive statistics of the different test forms, shape sets and distractor types, respectively. For an item-level breakdown of the response times by test form and shape set, see https://osf.io/g96f4/.

Finally, item dimensionality had a robust impact on performance, both in terms of accuracy ($\chi^2(1) = 1786.09$, $p < 0.001$) and response times ($\chi^2(1) = 2878.9$, $p < 0.001$). Specifically, item dimensionality linearly decreased the loglikelihood of responding correctly ($b = -0.38$, s.e. = 0.009), and increased response times ($b = 0.18$, s.e. = 0.003). Taken together, these results suggest that item properties that were not intended to affect performance (i.e. test forms, shape sets and distractor strategy) did not, whereas properties that were intended to modulate performance and response processes (i.e. item dimensionality) reliably did so.

## 3.2. Convergent validity

To assess convergent validity, we inspected the product–moment correlation between MaRs-IB performance (at the aggregate level) and matrix reasoning scores in the ICAR [29]. The correlation was 0.61 ($t(98) = 7.60$, $p < 0.001$, 95% CI = 0.47, 0.72) and a linear regression showed that it did not interact with the order in which participants did the two tasks ($p = 0.58$). Performance standard deviations of the ICAR matrix reasoning items in our sample and in Condon & Revelle [29] were, respectively, 0.27 and 0.5, possibly warranting correction for range restriction [29]. This resulted in a range-corrected correlation of 0.81, to be interpreted with caution, given that the assumptions for range restriction correction could not be fully tested. To assess the extent of divergent validity, we further compared the correlation described above to the correlations between the MaRs-IB and the remaining tasks of the ICAR (table 2), namely letter–number series completion, verbal reasoning and three-dimensional (3D) rotations. The highest correlation with MaRs-IB performance was indeed with the ICAR matrix reasoning (table 2). Furthermore, Fischer's *r–z* transform suggested that this correlation was higher than the correlation between the MaRs-IB and the 3D rotations ($p = 0.03$). However, it was not significantly different from the correlation between the MaRs-IB and letter–number series completion ($p = 0.05$), or between the MaRs-IB and the verbal reasoning task ($p = 0.11$).

Taken together, this evidence suggests that the MaRs-IB correlates strongly with the ICAR's matrix reasoning task, and more than with another non-verbal reasoning task of the ICAR. However, the uncorrected correlation size falls slightly short of the correlation size that could be expected from the positive manifold of fluid intelligence tests [29]. We also observed no clear evidence of divergent validity with regard to the ICAR's verbal reasoning task. We thus recommend further psychometric testing of our items, possibly using more established measures such as Raven's matrices [8] and larger samples.

We also inspected the relationship between MaRs-IB and digit span performance, as a measure of working memory, and found that the two measures were strongly correlated ($r = 0.45$, $t(657) = 12.74$, 95% CI [0.38, 0.5], $p < 0.001$). This is in line with previous meta-analyses, which have demonstrated an average correlation of $\hat{p} = 0.479$ between working memory and intelligence [3]. The relationship between MaRs-IB and digit span performance did not differ between age groups ($\chi^2(3) = 4.99$, $p = 0.172$).

**Table 2.** Correlations between performance on MaRs-IB items and tasks from the International Cognitive Ability Resource (MR, matrix reasoning; R3D, 3D rotations; LN, letter and number series completion; VR, verbal reasoning).

|       | ARTOL   | MR      | R3D     | LN      |
|-------|---------|---------|---------|---------|
| ARTOL |         |         |         |         |
| MR    | 0.61*** |         |         |         |
| R3D   | 0.45*** | 0.50*** |         |         |
| LN    | 0.44*** | 0.54*** | 0.45*** |         |
| VR    | 0.48*** | 0.47*** | 0.29*   | 0.45*** |

***$p < 0.001$, *$p < 0.05$. Bonferroni corrected.

**Table 3.** Descriptive statistics of the performance in the MaRs-IB by age group and gender. m, males; f, females. s.e. = standard error, IQR = inter-quartile range.

| age group | younger adolescents | mid-adolescents | older adolescents | adults |
|-----------|--------------------|-----------------|--------------------|--------|
| accuracy (mean) | 61 (m = 57, f = 63) | 68 (m = 67, f = 68) | 73 (m = 72, f = 73) | 81 (m = 79, f = 81) |
| accuracy (s.e.) | 1 (m = 2, f = 1) | 1 (m = 2, f = 2) | 1 (m = 2, f = 1) | 1 (m = 3, f = 1) |
| RT (median) | 6944 (m = 7006, f = 6872) | 7552 (m = 7964, f = 6841) | 7952 (m = 8774, f = 7582) | 9454 (m = 10 276, f = 9304) |
| RT (IQR) | 3879 (m = 4301, f = 3772) | 3723 (m = 4629, f = 2669) | 3309 (m = 3338, f = 2781) | 3231 (m = 3012, f = 3297) |
| items completed (mean) | 33.68 (m = 34.4, f = 33.27) | 32.29 (m = 32.17, f = 32.42) | 28.99 (m = 28.24, f = 29.52) | 33.43 (m = 31.96, f = 33.89) |
| items completed (s.e.) | 0.74 (m = 1.26, f = 0.92) | 0.76 (m = 1.19, f = 0.92) | 0.75 (m = 1.22, f = 0.94) | 0.93 (m = 2.07, f = 1.04) |
| inverse efficiency (median) | 11 838 (m = 12 655, f = 11 100) | 10 937 (m = 11 557, f = 10 150) | 11 057 (m = 12 285, f = 10 694) | 11 414 (m = 12 946, f = 11 226) |
| inverse efficiency (IQR) | 4031 (m = 4821, f = 3263) | 3852 (m = 4456, f = 3256) | 4023 (m = 3941, f = 4470) | 3668 (m = 4452, f = 3248) |

## 3.3. Age differences in performance

### 3.3.1. Accuracy

MaRs-IB accuracy differed between age groups ($\chi^2(3) = 10.39$, $p = 0.016$) (table 3). Planned contrasts suggested the age effect was mainly carried by adults performing significantly better than younger adolescents (figure 2; electronic supplementary material, table S6). The age effect was also evident when age was modelled as a continuous variable (figure 2). There was a significant, positive linear effect and a significant negative quadratic trend (table 4). Performance increased with age during adolescence before tapering off in adulthood (figure 2).

There was no significant effect of gender on accuracy ($\chi^2(1) = 3.77$, $p = 0.052$) (electronic supplementary material, figure S3) and no interaction between gender and age group ($\chi^2(3) = 0.68$, $p = 0.878$).

### 3.3.2. Response times

There was no significant effect of age group on response times ($\chi^2(3) = 1.54$, $p = 0.673$) (table 3) and none of the planned comparisons between age groups were significant (figure 3; electronic supplementary material, table S7). There was a slight positive linear trend, while the quadratic and cubic trends were non-significant (figure 3 and table 4). This indicates that there may be some increases in response times with age.

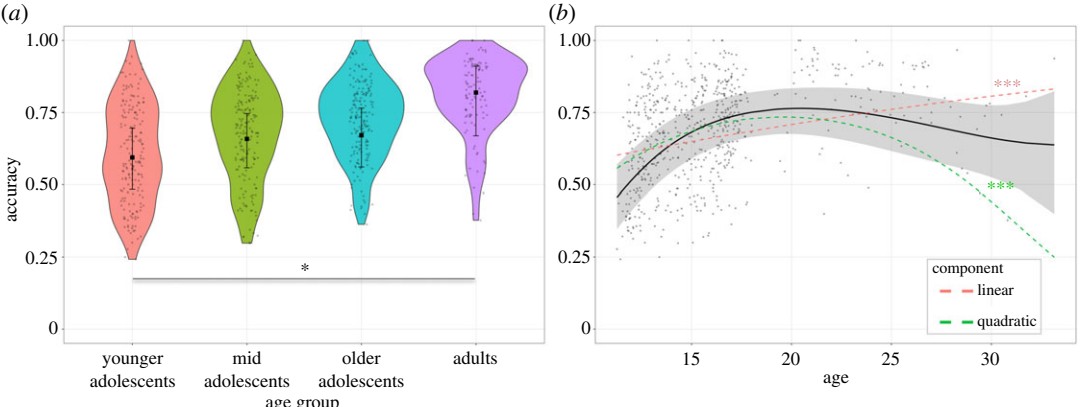

**Figure 2.** MaRs-IB accuracy by age group. Accuracy for each participant is shown. (*a*) Violin plots represent kernel probability density of the raw data at different values (randomly jittered across the *x*-axis). Within each age group, the black square represents the fixed effect estimate of accuracy from the GLMM, and error bars are the corresponding 95% CIs. See electronic supplementary material, table S6 for statistics of all contrasts. (*b*) Scatter plot of the relationship between age and accuracy in the MaRs-IB. The black line and shaded 95% CI show the overall polynomial trend. The coloured lines represent significant linear and quadratic trends. See table 4 for statistics of all trends. ***$p_{Bonf} < 0.001$, *$p_{Bonf.} < 0.05$.

**Table 4.** Polynomial trends for the effect of age on MaRs-IB performance.

| trend | *b* | *z* | *p*-value |
|---|---|---|---|
| **accuracy** | | | |
| linear | 33.14 | 4.05 | <0.001 |
| quadratic | −38.87 | −7.22 | <0.001 |
| cubic | 10.58 | 1.78 | 0.074 |
| **response times** | | | |
| linear | 5.15 | 2.43 | 0.015 |
| quadratic | −1.21 | −0.8 | 0.426 |
| cubic | −1.19 | −0.70 | 0.484 |
| **number of items completed** | | | |
| linear | −36.90 | −1.33 | 0.188 |
| quadratic | 29.17 | 2.12 | 0.035 |
| cubic | −7.28 | −0.54 | 0.589 |
| **inverse efficiency** | | | |
| linear | 5391.41 | 0.67 | 0.509 |
| quadratic | 9755.70 | 2.29 | 0.023 |
| cubic | −7861.30 | −1.86 | 0.064 |

There was a significant effect of gender on response times ($\chi^2(1) = 8.05$, $p = 0.005$) (electronic supplementary material, figure S3), with females responding faster than males (estimated mean$_{Females}$ = 5523 ms, s.e.$_{Females}$ = 466 ms, estimated mean$_{Males}$ = 5915 ms, s.e.$_{Males}$ = 502 ms). There was no significant interaction between gender and age ($\chi^2(3) = 3.58$, $p = 0.311$).

### 3.3.3. Number of items completed

The number of items completed did not differ between age groups ($\chi^2(3) = 0.76$, $p = 0.859$) (table 3). None of the age group comparisons were significant (figure 4; electronic supplementary material, table S8). When modelling age as a continuous variable there was no significant linear or cubic trend. There was, however, a significant quadratic trend of age, suggesting that the number of items

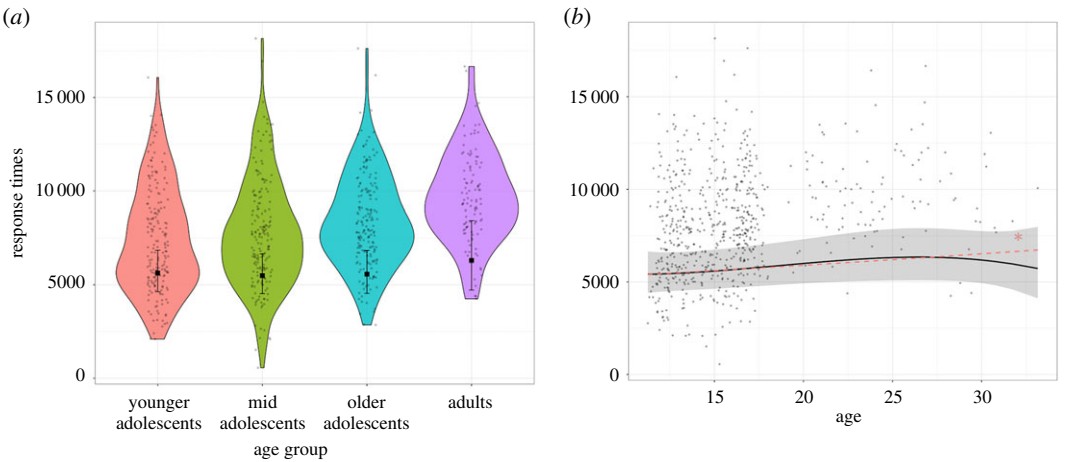

**Figure 3.** MaRs-IB response times by age group. (*a*) Violin plots of participants' raw median response times on correct items and age group-level fixed effects estimates of response times on correct items. None of the age group comparisons were significant. See electronic supplementary material, table S7 for statistics of all contrasts. (*b*) Scatter plot of the relationship between age and response times in the MaRs-IB. The black line and shaded 95% confidence interval show the overall polynomial trend. The red line represents the significant linear trend (table 4). *$p < 0.05$.

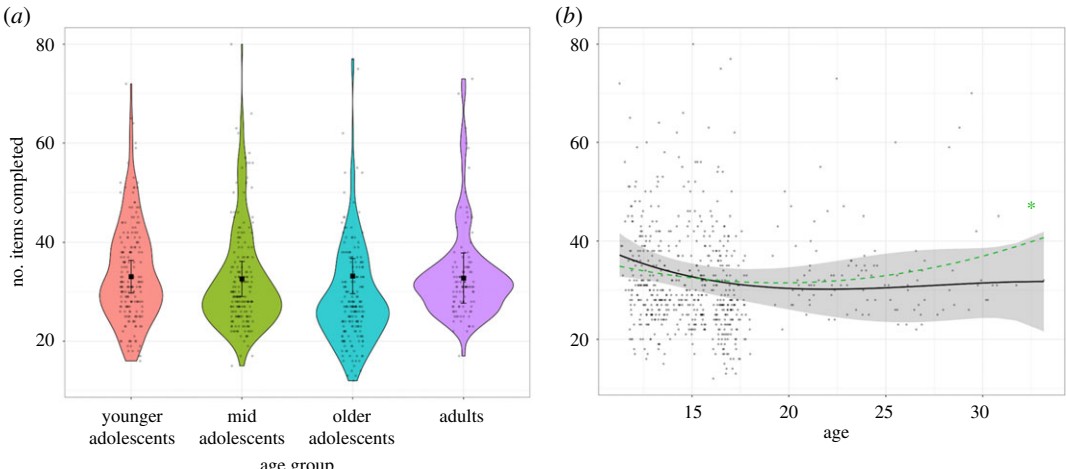

**Figure 4.** Number of MaRs-IB items completed by age group. (*a*) Violin plots of participants' raw number of items completed (correct and incorrect) and age group-level fixed effects estimates. None of the age group comparisons were significant. See electronic supplementary material, table S8 for statistics of all contrasts. (*b*) Scatter plot of the relationship between age and the number of items completed in the MaRs-IB. The black line and shaded 95% CI show the overall polynomial trend. The green line represents the significant quadratic trend (table 4). *$p < 0.05$.

completed was somewhat lower in late adolescence and early adulthood than before late adolescence or after early adulthood (figure 4). This is in line with the decrease in response times observed above and indicates a slight decrease in productivity during late adolescence and early adulthood.

There was no effect of gender on the number of items completed ($\chi^2(1) = 1.07$, $p = 0.300$), nor an interaction between gender and age ($\chi^2(3) = 2.96$, $p = 0.398$) (electronic supplementary material, figure S3).

### 3.3.4. Inverse efficiency

There was no significant main effect of age group on inverse efficiency ($\chi^2(3) = 3.76$, $p = 0.288$) (table 3) and none of the planned comparisons between age groups survived Bonferroni-correction (figure 5; electronic supplementary material, table S9). There was no linear or cubic effect of age on inverse efficiency (table 4). There was, however, a significant quadratic trend (figure 5 and table 4). This quadratic trend may indicate a peak in efficiency around late adolescence, consistent with the age-related nonlinear increases in accuracy and linear decreases in speed observed above.

There was a significant effect of gender on inverse efficiency ($\chi^2(1) = 24.45$, $p < 0.001$), with females responding more efficiently than males (estimated mean$_{Females}$ = 11222, estimated s.e.$_{Females}$ = 398;

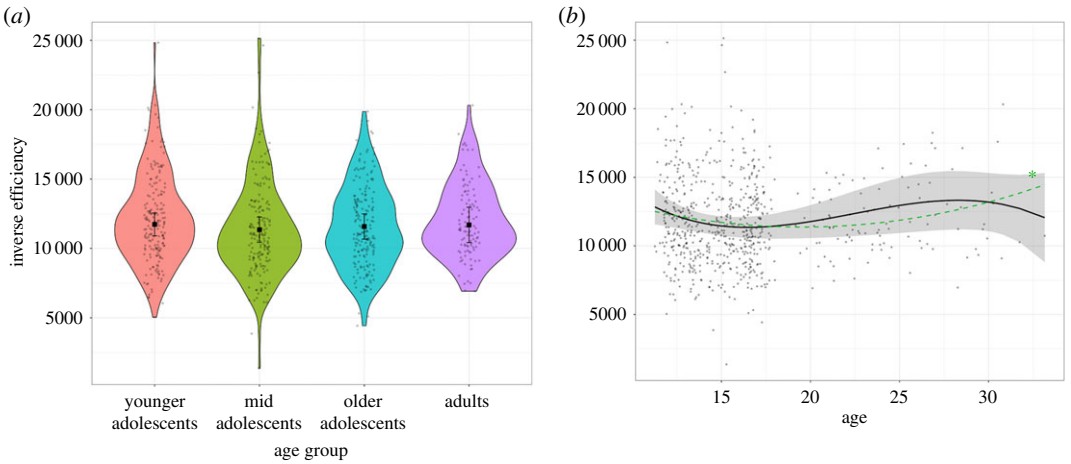

**Figure 5.** Abstract reasoning inverse efficiency by age group. (*a*) Violin plots of participants' raw inverse efficiency and age group-level fixed effects estimates. None of the age group comparisons were significant. See electronic supplementary material, table S9 for statistics of all contrasts. (*b*) Scatter plot of the relationship between age and inverse efficiency in the MaRs-IB. The black line and shaded 95% CI show the overall polynomial trend. The green line represents the significant quadratic trend (table 4). *$p < 0.05$.

estimated mean$_{\text{Males}}$ = 12522, estimated s.e.$_{\text{Males}}$ = 416.58) (electronic supplementary material, figure S3). This effect matched the pattern in accuracy and response times reported above: accuracy was comparable between genders but females responded faster. There was no interaction between gender and age group ($\chi^2(3) = 3.56$, $p = 0.314$).

## 4. Discussion

Here, we provide a large bank of novel, open-access abstract reasoning items that show acceptable internal consistency and convergent validity, and reasonable test–retest reliability. We propose that the MaRs-IB can be used for research purposes to complement existing, well-validated but copyrighted pen-and-paper tests [6,7], as well as computerized public-domain tests containing a small number of items tested in adults [12,13]. We provide a set of 80 items, as well as corresponding descriptive developmental data from adolescents and adults (aged 11–33 years). The MaRs-IB items can be implemented in computer-based experiments, allowing for flexible use ranging from online experiments to neuroimaging studies. The computerized implementation also allows for the measurement of response times, in addition to the standard accuracy measures. We further provide items in three shape variants to allow for repeated measures in longitudinal studies. We report performance and item-level statistics for different age groups to allow researchers to form item sets of custom difficulty and duration for developmental and non-developmental studies using information on accuracy, response times and efficiency.

The data analysed here suggest that accuracy in the MaRs-IB increases robustly until late adolescence, after which gains taper off. This is in line with previous research showing that reasoning capacity increases during childhood and adolescence, and into adulthood [21–23]. We extend the current literature by showing that response times increased slightly over the same developmental period and the number of items attempted decreased, resulting in a peak in efficiency in late adolescence. Overall, this suggests that accuracy increases past late-adolescence may be conveyed, in part, by response time slowing.

Two independent, but complementary lines of inquiry may provide insights into possible mechanisms of these developmental differences. First, developmental theories emphasize reductions in impulsivity over adolescence, which are linked to the protracted development of prefrontal regions over the same developmental period [26,27,39,40]. Second, computational theories of speed–accuracy trade-offs highlight that the time to arrive at a cognitive solution gives insights into the amount of evidence participants sample before deciding upon a solution [19,20]. The results obtained here suggest that participants increasingly sample evidence over the course of adolescence, resulting in more conservative decision thresholds at older ages. The peak in efficiency in late adolescence suggests that both accuracy increases and response time slowing drive developmental gains in abstract reasoning up until late adolescence, after which response time slowing may be a major driver of further improvements into adulthood. On a more general level, this finding highlights the value of collecting response time, as well as accuracy data, for providing insights into the developmental mechanisms of abstract reasoning.

We believe that the advantages of providing items open-access, including flexibility of use and equity of access, ultimately outweigh potential costs of open-access provision. Such costs include the possibility that participants may become familiar with or train on the items. This problem is not unique to open-access items—participants are already able to access many matrix reasoning tasks if they wish to. For now, some of the gold-standard abstract reasoning tests, which have been used for decades, also arguably carry a higher likelihood of familiarity than novel items like MaRs-IB. We cannot rule out the possibility that this might change in the long run, however. To address this issue, researchers may wish to implement pre- and post-screening procedures to identify (and possibly exclude) participants familiar with the MaRs-IB. The post-screening procedure may include using outlier detection methods such as examining person-fit statistics [41] or applying DIF analyses to identify participants behaving unusually on MaRs-IB items [42].

Despite strengths of the MaRs-IB, including the large number of items and developmental data, we note several limitations. First, we observed gender differences in response times and efficiency, but not accuracy, with females responding faster than males, resulting in overall higher levels of efficiency. We hypothesize that this effect may be due to low-level perceptual properties of the stimuli, particularly colour-choices. Females generally show more acute colour discrimination [43] and are less likely to present with colour-blindness than males [44]. This may have resulted in faster response times overall. To allow researchers to address colour-decimation as a potential confound in future studies, we provide a separate set of colour vision deficiency-friendly stimuli here: https://osf.io/g96f4/. Second, the data analysed here were originally collected as part of a cognitive training study [28] and were therefore not optimized for psychometric validation. In particular, we did not have enough observations to assess the psychometric properties of some of the later items of the MaRs-IB (which few participants reached). While we were able to provide evidence for acceptable internal consistency in a subset of earlier items that involved no variability in completion rate, further psychometric validation of the MaRs-IB is recommended. We also do not have data on our sample's distribution and representativeness in terms of demographic variables such as socio-economic status and ethnicity, and particular age brackets (e.g. between 18 and 20) are under-sampled. We therefore recommend a more extensive validation of these items with a representative sample in the future. While we encourage the use of the items in measuring non-verbal reasoning in development samples, our item bank should not be considered a normative measure of intelligence until further psychometric testing has been completed.

## 5. Conclusion

Complementing existing, mostly copyrighted, reasoning tasks, we here present a large set of abstract reasoning items which can be used and adapted, free-of-charge, for computerized research studies requiring large sets of stimuli. The MaRs-IB is sensitive to age differences in accuracy, which suggests that further psychometric validation will make the item bank a useful resource for researchers interested in abstract reasoning.

Ethics. All participants provided informed consent and all procedures were approved by UCL's ethical committee (Project ID: 4870/001; Project title: 'Development of cognitive processing during adolescence').

Data accessibility. Working towards an open science framework, the study in question provides 80 items (each in three variants) for a novel matrix reasoning item bank, a ready-to-use online implementation and descriptive data from 659 participants aged between 11 and 33 years. The manuscript is already available in preprint (https://osf.io/uvteh/). In an open repository (https://osf.io/g96f4/), we also provide the novel items and item-level data. Owing to the limitations of the ethical approvals that were granted some years in advance of this research being conducted, the authors regret that the full datasets cannot currently be made publicly available. To do so would risk revealing data that may contribute to individuals in the study (currently anonymized) becoming identifiable. Specifically, we have not made available the participant-level data. The participant-level data (and code) will be made available upon reasonable request.

Authors' contributions. S.-J.B. wrote the initial grant application and protocol. S.-J.B., D.F., L.J.K. and A.L.S. developed the original study design. Data collection for the original study were performed by D.F., L.J.K. and A.L.S. G.C. and S.-J.B. designed the validation study. Data collection for the validation study was performed by B.P.P.-S. and G.C. G.C. and D.F. performed the data analyses and interpretation, under the supervision of S.-J.B. G.C. and D.F. contributed equally to the manuscript. All authors approved the final version of the manuscript for submission.

Competing interests. The authors declare no conflict of interest.

Funding. The study was funded by the Royal Society, the Jacobs Foundation, the Nuffield Foundation, and Wellcome Trust (grant no. 104908/Z/14/Z). D.F. was supported by the Cusanuswerk, University College London and the UK Medical Research Council.

**Acknowledgements.** The authors would like to thank all participants and participating schools, as well as Alberto Lazari, Somya Iqbal and Fabian Stamp for help with data collection. The authors would also like to thank Yaara Erez and Rogier Kievit for helpful discussions about the items.

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
