## [Reviewer comments · Royal Society Open Science]

Review History

RSOS-190232.R0 (Original submission)

Review form: Reviewer 1

Is the manuscript scientifically sound in its present form?

No

Are the interpretations and conclusions justified by the results?

No

Is the language acceptable?

Yes

Is it clear how to access all supporting data?

Yes

Do you have any ethical concerns with this paper?

No

Have you any concerns about statistical analyses in this paper?

Yes

Recommendation?

Major revision is needed (please make suggestions in comments)

Comments to the Author(s)

See attached file (Appendix A).

Review form: Reviewer 2

Is the manuscript scientifically sound in its present form?

Yes

Are the interpretations and conclusions justified by the results?

No

Is the language acceptable?

Yes

Is it clear how to access all supporting data?

Yes

Do you have any ethical concerns with this paper?

No

Have you any concerns about statistical analyses in this paper?

No

Recommendation?

Major revision is needed (please make suggestions in comments)

Comments to the Author(s)

This is an interesting dataset that can potentially provide researchers with a useful computerised reasoning task for use in future studies. However, I think that conclusions regarding the utility of this novel task may be premature based on the data presented in the manuscript.

1) A comparison with a reasoning measure (e.g., Raven's Matrices) is missing. This is a critical limitation to using the ART as an alternative measure of reasoning abilities without further investigation. This should be discussed in the paper, and I think language in the introduction and discussion should be revised as not to overstate the utility of the task at the present stage.

2) I am not convinced that the ART task shows good sensitivity to different age-related differences in reasoning. The results regarding accuracy showed that age effects were driven primarily by the difference between adults and younger adolescents, whereas the task is not really sensitive to age differences between mid and older adolescents. In addition, the measures incorporating response time information (which I consider a critical advantage of the ART) such

as the inverse efficiency did not show stable differences, apart from more subtle effects when age was included as a continuous variable. In the absence of an established reasoning measure to evaluate these results against, I find this claim unjustified.

3) The separation in different age groups was not very clear - whereas the adolescent groups cover a more or less similar age range of 2 - 3 years, the adult includes participants between 18 - 33 years. Given that reasoning development may not be completed by age 18, and most cognitive abilities start declining before age 33 years, it is difficult to use this group as a reference. Looking at the figures, it seems participants in the adult group were not sampled homogeneously across the age range, with only few participants between 18-20 years and > 30 years.

Decision letter (RSOS-190232.R0)

24-Apr-2019

Dear Dr Chierchia,

The editors assigned to your paper ("The Abstract Reasoning Task (ART): Normative Data for a Novel, Open-Access Abstract Reasoning Task in a Sample of Adolescents and Adults") have now received comments from reviewers. We would like you to revise your paper in accordance with the referee and Associate Editor suggestions which can be found below (not including confidential reports to the Editor). Please note this decision does not guarantee eventual acceptance.

Please submit a copy of your revised paper before 17-May-2019. Please note that the revision deadline will expire at 00.00am on this date. If we do not hear from you within this time then it will be assumed that the paper has been withdrawn. In exceptional circumstances, extensions may be possible if agreed with the Editorial Office in advance. We do not allow multiple rounds of revision so we urge you to make every effort to fully address all of the comments at this stage. If deemed necessary by the Editors, your manuscript will be sent back to one or more of the original reviewers for assessment. If the original reviewers are not available, we may invite new reviewers.

If your study uses humans or animals please include details of the ethical approval received, including the name of the committee that granted approval. For human studies please also detail

whether informed consent was obtained. For field studies on animals please include details of all permissions, licences and/or approvals granted to carry out the fieldwork.

- Data accessibility

If you wish to submit your supporting data or code to Dryad (<http://datadryad.org/>), or modify your current submission to dryad, please use the following link:
<http://datadryad.org/submit?journalID=RSOS&manu=RSOS-190232>

- Competing interests

- Authors' contributions

- Acknowledgements

- Funding statement

Kind regards,
Andrew Dunn
Royal Society Open Science Editorial Office

on behalf of Dr Joydeep Bhattacharya (Associate Editor) and Essi Viding (Subject Editor)
openscience@royalsociety.org

Associate Editor's comments (Dr Joydeep Bhattacharya):

Although both reviewers have found merit and interest in your submission, both have expressed significant concerns. Reviewer 1 is especially concerned about the validity of your conclusion and also asked for further clarity on the reliability of the proposed test. Do note that a copyrighted test with an identical title is available in the US. Reviewer 2 has suggested to include a comparison with a standard reasoning measure, and further, questioned about the sensitivity of the proposed test battery.

Comments to Author:

Reviewers' Comments to Author:

Reviewer: 1

Comments to the Author(s)

See attached file

Reviewer: 2

Comments to the Author(s)

This is an interesting dataset that can potentially provide researchers with a useful computerised reasoning task for use in future studies. However, I think that conclusions regarding the utility of this novel task may be premature based on the data presented in the manuscript.

1) A comparison with a reasoning measure (e.g., Raven's Matrices) is missing. This is a critical limitation to using the ART as an alternative measure of reasoning abilities without further investigation. This should be discussed in the paper, and I think language in the introduction and discussion should be revised as not to overstate the utility of the task at the present stage.

2) I am not convinced that the ART task shows good sensitivity to different age-related differences in reasoning. The results regarding accuracy showed that age effects were driven primarily by the difference between adults and younger adolescents, whereas the task is not really sensitive to age differences between mid and older adolescents. In addition, the measures incorporating response time information (which I consider a critical advantage of the ART) such as the inverse efficiency did not show stable differences, apart from more subtle effects when age was included as a continuous variable. In the absence of an established reasoning measure to evaluate these results against, I find this claim unjustified.

3) The separation in different age groups was not very clear - whereas the adolescent groups cover a more or less similar age range of 2 - 3 years, the adult includes participants between 18 - 33 years. Given that reasoning development may not be completed by age 18, and most cognitive abilities start declining before age 33 years, it is difficult to use this group as a reference. Looking at the figures, it seems participants in the adult group were not sampled homogeneously across the age range, with only few participants between 18-20 years and > 30 years.

Author's Response to Decision Letter for (RSOS-190232.R0)

See Appendix B.

RSOS-190232.R1 (Revision)

Review form: Reviewer 1

Is the manuscript scientifically sound in its present form?

No

Are the interpretations and conclusions justified by the results?

No

Is the language acceptable?

No

Is it clear how to access all supporting data?

Yes

Do you have any ethical concerns with this paper?

No

Have you any concerns about statistical analyses in this paper?

Yes

Recommendation?

Reject

Comments to the Author(s)

Staff note: Please see the attached file for detailed comments to the authors (Appendix C).

Review form: Reviewer 2

Is the manuscript scientifically sound in its present form?

Yes

Are the interpretations and conclusions justified by the results?

Yes

Is the language acceptable?

Yes

Is it clear how to access all supporting data?

Yes

Do you have any ethical concerns with this paper?

No

Have you any concerns about statistical analyses in this paper?

No

Recommendation?

Accept with minor revision (please list in comments)

Comments to the Author(s)

The authors have fully addressed my concerns. I would recommend to remove the few remaining places in the text and in the data sharing statement that refer to the presented data as normative.

Decision letter (RSOS-190232.R1)

17-Jul-2019

Dear Dr Chierchia:

Manuscript ID RSOS-190232.R1 entitled "The Abstract Reasoning Task of London (ARTOL): A Novel, Open-Access Abstract Reasoning Task for Adolescents and Adults" which you submitted to Royal Society Open Science, has been reviewed. The comments of the reviewer(s) are included at the bottom of this letter.

Please submit a copy of your revised paper before 09-Aug-2019. Please note that the revision deadline will expire at 00.00am on this date. If we do not hear from you within this time then it will be assumed that the paper has been withdrawn. In exceptional circumstances, extensions may be possible if agreed with the Editorial Office in advance. We do not typically allow multiple rounds of revision so we urge you to make every effort to fully address Reviewer 1 comments. If deemed necessary by the Editors, your manuscript will be sent back to one or more of the original reviewers for assessment. If the original reviewers are not available we may invite new reviewers.

- Ethics statement

If your study uses humans or animals please include details of the ethical approval received, including the name of the committee that granted approval. For human studies please also detail

whether informed consent was obtained. For field studies on animals please include details of all permissions, licences and/or approvals granted to carry out the fieldwork.

- Data accessibility

- Competing interests

- Authors' contributions

- Acknowledgements

- Funding statement

Kind regards,

on behalf of Dr Joydeep Bhattacharya (Associate Editor) and Essi Viding (Subject Editor)
openscience@royalsociety.org

Reviewer comments to Author:

Reviewer: 1

Comments to the Author(s)

Staff note: Please see the attached file for detailed comments to the authors

Reviewer: 2

Comments to the Author(s)

The authors have fully addressed my concerns. I would recommend to remove the few remaining places in the text and in the data sharing statement that refer to the presented data as normative.

Author's Response to Decision Letter for (RSOS-190232.R1)

See Appendix D.

RSOS-190232.R2 (Revision)

Review form: Reviewer 1

Is the manuscript scientifically sound in its present form?

Yes

Are the interpretations and conclusions justified by the results?

Yes

Is the language acceptable?

No

Do you have any ethical concerns with this paper?

No

Have you any concerns about statistical analyses in this paper?

No

Recommendation?

Accept with minor revision (please list in comments)

Comments to the Author(s)

A file is attached (Appendix E).

Decision letter (RSOS-190232.R2)

18-Sep-2019

Dear Dr Chierchia,

On behalf of the Editors, I am pleased to inform you that your Manuscript RSOS-190232.R2 entitled "The Matrix Reasoning Item Bank (MaRs-IB): Novel, Open-Access Abstract Reasoning Items for Adolescents and Adults" has been accepted for publication in Royal Society Open Science subject to minor revision in accordance with the referee suggestions. Please find the referees' comments at the end of this email.

The reviewers and Subject Editor have recommended publication, but also suggest some minor revisions to your manuscript. Therefore, I invite you to respond to the comments and revise your manuscript.

- Ethics statement

- Data accessibility

<http://datadryad.org/submit?journalID=RSOS&manu=RSOS-190232.R2>

- Competing interests

- Authors' contributions

- Acknowledgements

- Funding statement

Because the schedule for publication is very tight, it is a condition of publication that you submit the revised version of your manuscript before 27-Sep-2019. Please note that the revision deadline will expire at 00.00am on this date. If you do not think you will be able to meet this date please let me know immediately.

Kind regards,

Royal Society Open Science Editorial Office
Royal Society Open Science
openscience@royalsociety.org

on behalf of Dr Joydeep Bhattacharya (Associate Editor) and Essi Viding (Subject Editor)
openscience@royalsociety.org

Reviewer comments to Author:

Reviewer: 1

Please see the attached file.

Author's Response to Decision Letter for (RSOS-190232.R2)

See Appendix F.

Decision letter (RSOS-190232.R3)

23-Sep-2019

Dear Dr Chierchia,

I am pleased to inform you that your manuscript entitled "The Matrix Reasoning Item Bank (MaRs-IB): Novel, Open-Access Abstract Reasoning Items for Adolescents and Adults" is now accepted for publication in Royal Society Open Science.

on behalf of Dr Joydeep Bhattacharya (Associate Editor) and Essi Viding (Subject Editor)
openscience@royalsociety.org

Appendix A

Review: “The Abstract Reasoning Task (ART): Normative Data for a Novel, Open-Access Abstract Reasoning Task in a Sample of Adolescents and Adults”

Summary

The goal of this study is to provide psychometric support for a reasoning test that is freely available for research use. The *Standards for Educational and Psychological Tests* (2014) include many psychometric specifications for reliability, validity and norms. Although the item design is well specified and equated over test forms in the current study, the results presented do not include many fundamental psychometric analyses to evaluate overall test quality, item properties and test form comparability. Also, the current study cannot be deemed normative, as no descriptive statistics or score correspondence between test forms are provided and it is not clear the sample of participants would be sufficiently representative to qualify for normative based scaling. Finally, the title of the measure, Abstract Reasoning Task (ART) is too similar to a copyrighted matrix test with more established psychometric properties and scaling (i.e., Abstract Reasoning **Test** (ART)). It is suggested that another name and acronym be used for this test (e.g., Matrix Reasoning Task (MRT) might not conflict with other tests).

Specific Comments:

1. P. 10 Test design: counter-balancing item properties is a good strategy. However, what is the predictability of item difficulty from the various design factors? That is evidence for the response processes aspect of validity.
2. P.11 Item analysis, from a classical test perspective, should include p-values and item total correlations (biserial r). Nothing is presented.
3. P. 11 The goal for using GLMM is not clear. This is not the typical way to handle item analysis and obviously it was not feasible. Less information about GLMM is preferable, except that it was not appropriate. Further, age and gender effects psychometrically should be considered in the context of differential item functioning....a very different analysis than presented in text.
4. P.12 the log link for RT data is fairly standard and the inverse link function need not be considered.
5. P. 13 How is scoring conducted for items that are not completed? Scored as 0? Unfortunately, classical test theory approaches as applied in the study (versus contemporary item response theory) do not have a good solution.
6. P. 13 Split-half reliability, given SB correction, is only moderate. Further, split-half reliability is NOT internal consistency (e.g., KR-20 or Cronbach Alpha). Given the moderate split-half reliability, internal consistency may not be sufficiently high.
7. P 14 ? Parallel forms reliability was assessed by SB on P. 13. It is unclear how the analyses are being conducted. Raw data or item means?
8. P. 16 Of course, age differences are expected. However, although the figures are somewhat informative, a table of descriptive statistics would be helpful for all dependent variables.

9. P. 23 The limitations section definitely should be included in the paper. In fact, the split-half reliabilities are relatively low and should also be included in limitations.

Appendix B

Reviewer 1

Summary

The goal of this study is to provide psychometric support for a reasoning test that is freely available for research use. The Standards for Educational and Psychological Tests (2014) include many psychometric specifications for reliability, validity and norms. Although the item design is well specified and equated over test forms in the current study, the results presented do not include many fundamental psychometric analyses to evaluate overall test quality, item properties and test form comparability. Also, the current study cannot be deemed normative, as no descriptive statistics or score correspondence between test forms are provided and it is not clear the sample of participants would be sufficiently representative to qualify for normative based scaling. Finally, the title of the measure, Abstract Reasoning Task (ART) is too similar to a copyrighted matrix test with more established psychometric properties and scaling (i.e., Abstract Reasoning Test (ART)). It is suggested that another name and acronym be used for this test (e.g., Matrix Reasoning Task (MRT) might not conflict with other tests).

We thank the reviewer for these comments. We agree that our sample of participants may not be sufficiently representative to qualify for normative based scaling. We have thus removed any reference to the data as 'normative'. We have also added the psychometric analyses recommended by the reviewer (please see our specific replies below) and have changed the name of the task to the Abstract Reasoning Task of London or "ARTOL". To facilitate referencing we have also copied each of the novel sections of the manuscript below, under the corresponding replies.

Specific Comments:

1. *P. 10 Test design: counter-balancing item properties is a good strategy. However, what is the predictability of item difficulty from the various design factors? That is evidence for the response processes aspect of validity.*

We have followed the reviewer's suggestion and ran new analyses relating item difficulty to performance and response times. The latter can also provide insight into the response processes aspect of validity (*The Standards for Educational and Psychological Tests, 2014*). In synthesis, the new analyses suggest that item difficulty (which is proportional to the number of dimensions that are changing in a puzzle) reliably predict accuracy (increased item difficulty is correlated with decreased accuracy) and response times (increased item difficulty is correlated with increased RTs). Taken together with the existing sections, these results suggest that task properties that were not intended to affect performance (i.e., puzzle sets, shape sets and distractor strategy) did not, whereas properties that were intended to modulate performance (i.e., item difficulty) did so (see p. 14 and 17 of the revised manuscript, copied below).

"To assess parallel-forms reliability and response processes of the ARTOL we used exploratory GLMMs specified as described above to investigate the effect of distractor type (minimal or paired difference), puzzle set (puzzle set 1, 2 or 3), shape set (shape set 1, 2 or 3) and item difficulty (a score ranging between 1 and 8, reflecting the number of dimensions changing in a puzzle) on accuracy and

response times. For the analysis of difficulty only, the item-related random intercept was removed because of multicollinearity between each item and the associated difficulty” (p. 12-13).

“Finally, item difficulty had a robust impact on performance, both in terms of accuracy ($\chi^2(1) = 1786.09, p < 0.001$) and response times ($\chi^2(1) = 2878.8, p < 0.001$). Specifically, item difficulty linearly decreased the loglikelihood of responding correctly ($b = -0.38, SE = 0.008$), and increased response times ($b = 0.17, SE = 0.003$). Taken together, these results suggest that task properties that were not intended to affect performance (i.e., puzzle sets, shape sets and distractor strategy) did not, whereas properties that were intended to modulate performance and response processes (i.e., item difficulty) reliably did so.” (p. 17-18)

2. *P.11 Item analysis, from a classical test perspective, should include p-values and item total correlations (biserial r). Nothing is presented.*

We thank the reviewer for this suggestion. Point biserial correlations between each item and mean performance on the remaining items, as well p-values are now presented in the manuscript, as suggested by the reviewer:

“[...] focusing on items for which we had at least 40 observations (given that relatively few participants reached the later items of the task). We also excluded the first 5 items, which were deliberately easier to familiarize participants with the task and had virtually no variance (see task description). This analysis focused on 46 items [...]. The mean item-total biserial correlation for the same items was 0.31 ($SE = 0.01$) (biserial correlation of each item is available in Supplementary Table S9), thus in acceptable range (Everitt & Skrondal, 2002), with a mean item p-value (i.e., the proportion of participants that answered correctly to each item) equal to 0.54 ($SE = 0.03$) (p-values of each item are available in the Supplementary Table S9).” (p. 14)

3. *P. 11 The goal for using GLMM is not clear. This is not the typical way to handle item analysis and obviously it was not feasible. Less information about GLMM is preferable, except that it was not appropriate. Further, age and gender effects psychometrically should be considered in the context of differential item functioning....a very different analysis than presented in text.*

GLMMs was our method of choice because GLMMs are a versatile and rigorous analysis method that has become increasingly common in Psychological research dealing with random effects such as subject-level variability (Bolker et al., 2009; Zuur, 2009). Standard regression approaches assume “independently distributed error terms for the individual observations within a sample” (Finch, Bolin, & Kelley, 2014), an assumption that is clearly violated our data. In the same vein, “students in a classroom” (or in a school in our case) has explicitly been mentioned as an example in which mixed models are the analysis method of choice (West, Welch, & Galecki, 2004). Using GLMMs to control for item-level variability in the same manner as subject- or school-level variability seems an

entirely plausible and useful extension. Further advantages of GLMMs include their ability to deal with unbalanced datasets, which are typical in datasets such as ours. These considerations suggest that GLMMs are in fact a useful analysis approach to our data, allowing us to run analyses at the trial-level, grouping error terms by items and participant (the latter nested within schools). We now are more explicit in stating the goal of applying GLMMs to our data (p. 13, copied below).

“We modelled each of these four dependent variables using mixed models because these allowed to accurately partition the error terms according to the hierarchical structure of the data (e.g., trial-level data was clustered by participants, which were in turn nested in schools) (Bates, Maechler, & Bolker, 2013).” (p. 12)

Furthermore, following the reviewer’s advice that less information is preferable, we have removed information regarding the selection procedure for the random effect terms (p. 12-13).

We have also followed the reviewer’s advice and run new analyses investigating differential item functioning (DIF). We did so using logistic regression (Swaminathan & Rogers, 1990). As targets for potential DIF we focused on age (both as categorical and continuous) and gender. As for the biserial correlation analysis described above, this analysis was not possible for several items due to there being too few observations in each age group. However, of the same subset of 46 items discussed above, none displayed DIF, suggesting that they are unbiased relative to age and gender. The novel sections are copied below.

“To investigate possible differential item functioning (“DIF”) we used logistic regression (Swaminathan & Rogers, 1990), as implemented in the “difR” package in R (Magis, Béland, Tuerlinckx, & De Boeck, 2010). We tested for both uniform and non-uniform DIF, correcting for multiple comparisons using the Benjamini-Hochberg method, as recommended by Kim & Oshima (2013). As potential sources of differential item functioning, we focused on age (used as both categorical and continuous) and gender.” (p. 13)

“None of these items displayed uniform or non-uniform differential item functioning, suggesting they are unbiased relative to the age groups tested here and relative to gender.” (p. 14).

Finally, to alleviate any further concerns with regards to the use of GLMMs, we re-ran all analyses aggregating data at the participant level and running standard regression analyses using only the main factor of interest (i.e., age) as independent variable. These models corroborated all of our main findings: there was a main effect of age on accuracy ($F(3,655) = 43.23, p < 0.001$) and response times ($F(3,655) = 17.38, p < 0.001$), with linear and quadratic trends when using age as a continuous variable (accuracy: linear trend = 1.56, $SE = 0.15, F(1,656) = 107.64, p < 0.001$; quadratic trend = -1.00, $SE = 0.15, F(1,656) = 43.80, p < 0.001$; response times: linear trend = 19907, $SE = 2714, F(1,656) = 53.78, p < 0.001$; quadratic trend = - 6960, $SE = 2714, F(1,656) = 6.58, p < 0.05$), all of

which in the same direction as predicted by the GLMMs. There was also a main effect of age on the number of trials completed ($F(3,655) = 7.97, p < 0.001$). Similarly to the GLMM results, there was no linear effect of age on the number of trials completed ($p = 0.754$), but a significant quadratic trend when using age as a continuous variable (quadratic trend = 37.49, $SE = 10.1, F(1,655) = 13.78, p < 0.001$). Finally, as for the GLMM results, there was no main effect of age on inverse efficiency ($p = 0.09$), but a significant quadratic trend of age (quadratic trend = 7286, $SE = 3140, F(1,655) = 5.38, p < 0.05$) corroborating the GLMM result that efficiency peaked during adolescence in our data. In synthesis, our GLMM results can be replicated using more traditional analysis approaches.

4. *P.12 the log link for RT data is fairly standard and the inverse link function need not be considered.*

We have removed the section considering the inverse link function, as suggested (p. 13).

5. *P. 13 How is scoring conducted for items that are not completed? Scored as 0? Unfortunately, classical test theory approaches as applied in the study (versus contemporary item response theory) do not have a good solution.*

Performance was measured at the trial level after removal of incomplete trials. This is now explicitly stated (page 12).

6. *P. 13 Split-half reliability, given SB correction, is only moderate. Further, split-half reliability is NOT internal consistency (e.g., KR-20 or Cronbach Alpha). Given the moderate split-half reliability, internal consistency may not be sufficiently high.*

We have followed the reviewer's suggestion and no longer describe SB as a measure of internal consistency. We now describe it as a reliability measure (Eisinga, Grotenhuis & Pelzer, 2013). Moreover, we computed KR-20 (focusing on the item for which we had at least 40 observations and excluding the first 5 trials, as also done above). This resulted in a KR-20 of 0.74, thus in the acceptable range (Nunnally, 1978). This analysis is now described on p. 16. The novel section is copied below.

"To assess internal consistency we computed the Kuder-Richardson 20 formula focusing on [the subset of 46 items described above, which] resulted in a KR-20 of 0.74 (95th CI = [0.72 0.76])." (p. 14)

7. *P 14 ? Parallel forms reliability was assessed by SB on P. 13. It is unclear how the analyses are being conducted. Raw data or item means?*

SB was assessed using trial-level raw data, this is now clearly stated (p. 15).

8. P. 16 Of course, age differences are expected. However, although the figures are somewhat informative, a table of descriptive statistics would be helpful for all dependent variables.

This has been added (see Table 2 in the MS, p. 19, also copied below).

Table 2. Descriptive Statistics of performance in the ARTOL task by Age group and Gender. m = males, f = females.

Age group	Younger adolescents	Mid adolescents	Older adolescents	Adults
Accuracy (mean)	61 (m=57 f=63)	68 (m=67 f=68)	73 (m=72 f=73)	81 (m=79 f=81)
Accuracy (SE)	33.68 (m=34.4 f=33.27)	32.29 (m=32.17 f=32.42)	28.99 (m=28.24 f=29.52)	33.43 (m=31.96 f=33.89)
RT (median)	6944 (m=7006 f=6872)	7552 (m=7964 f=6841)	7952 (m=8774 f=7582)	9454 (m=10276 f=9304)
RT (IQR)	3879 (m=4301 f=3772)	3723 (m=4629 f=2669)	3309 (m=3338 f=2781)	3231 (m=3012 f=3297)
Trials completed (mean)	24 (m=24 f=25)	30 (m=30 f=32)	36 (m=36 f=42)	38 (m=38 f=40)
Trials completed (SE)	16 (m=17 f=16)	15 (m=15 f=17)	12 (m=13 f=12)	17 (m=17 f=22)
Inverse efficiency (median)	11838 (m=12655 f=11100)	10937 (m=11557 f=10150)	11057 (m=12285 f=10694)	11414 (m=12946 f=11226)
Inverse efficiency (IQR)	4031 (m=4821 f=3263)	3852 (m=4456 f=3256)	4023 (m=3941 f=4470)	3668 (m=4452 f=3248)

9. P. 23 *The limitations section definitely should be included in the paper. In fact, the split- half reliabilities are relatively low and should also be included in limitations.*

In the limitation section, we now acknowledge that the sample may not be representative enough to allow for normative use, that split-half reliabilities are moderate (as the reviewer suggests in point 6) and that some age ranges are under-represented (see Reviewer 2's point 3). We also recommend that researchers select items with highest biserial item-total correlation when customizing the task to their needs. This item level information is available in the Supplementary Material (Supplementary Table S9). The revised limitations section is copied below.

“Second, the data analysed here was originally collected as part of a cognitive training study (Knoll et al., 2016) and was therefore not optimized for psychometric validation. While we were able to provide evidence for moderate

internal consistency and reasonable test-retest reliability and convergent validity, we did not have enough observations to assess the psychometric properties of some of the later items of the task (which few participants reached). We recommend that researchers select items with highest biserial item-total correlation when customizing the task to their needs. This information is available in the Supplementary Material (Table S9). We also do not have data on our sample's distribution and representativeness in terms of demographic variables such as socio-economic status and ethnicity, and particular age brackets (e.g., between 18 and 20) are under-sampled. We therefore recommend a more extensive validation of this task with a representative sample in the future. Finally, while we observe an acceptable degree of convergent validity between the ARTOL and the matrix reasoning items of the ICAR, we observe no significant evidence of divergent validity from other ICAR measures. Because of these limitations, we do not advocate the use of this task as a normative measure of IQ, but instead view it purely as a measure of abstract reasoning to be used for research purposes only." (p. 26-27)

Reviewer 2

Comments to the Author(s)

This is an interesting dataset that can potentially provide researchers with a useful computerised reasoning task for use in future studies. However, I think that conclusions regarding the utility of this novel task may be premature based on the data presented in the manuscript.

- 1) A comparison with a reasoning measure (e.g., Raven's Matrices) is missing. This is a critical limitation to using the ART as an alternative measure of reasoning abilities without further investigation. This should be discussed in the paper, and I think language in the introduction and discussion should be revised as not to overstate the utility of the task at the present stage.*

This is a really important point, and we have followed the reviewer's suggestion and collected new data in order to compare our task to a more validated and standard task involving matrix reasoning the "International Cognitive Ability Resource" (ICAR; Condon & Revelle, 2014). The raw correlation between ARTOL performance and the ICAR matrix reasoning component was 0.54 ($p < 0.001$), thus in acceptable range for convergent validity purposes (Carlson & Herdman, 2012). The correlation was 0.79 when correcting for range restriction (Condon & Revelle, 2014). The novel sections of the manuscript are copied below.

"A follow-up study was conducted to assess convergent validity of the ARTOL. A power analysis suggested that 38 participants are sufficient to detect a correlation of 0.5 at 90% power. For this study, we thus recruited 50 participants (36 females, 14 males, mean age = 24.18, $SE = 0.49$, age range 20-35 years)." (p. 7).

"In the follow-up study, we asked participants to complete the ARTOL, as well a more established task: the "International Cognitive Ability Resource" (ICAR)

(Condon & Revelle, 2014). The latter includes four main tasks: matrix reasoning, series completion, spatial rotations and verbal reasoning. We particularly focused on the matrix reasoning items of the ICAR, as these most closely resembled the ARTOL items and other tests like Raven’s Progressive matrices. The ARTOL and ICAR were administered in counterbalanced order.” (p. 11)

“To investigate convergent validity in the follow-up study, we inspected the product moment correlation between performance in the ARTOL and the matrix reasoning items of the ICAR. Following Condon & Revelle (2014), we further assessed the presence of possible range restriction by comparing the standard deviations of the matrix reasoning items in our sample and theirs. We obtained the latter by combining the standard deviations for the ICAR matrix reasoning items (listed in Table 2 of p. 55). Range correction was performed with the “rangeCorrection” function of the Psych package in R (Revelle, 2018).” (p. 13)

“To assess convergent validity we inspected the product-moment correlation between ARTOL performance (at the aggregate level) and matrix reasoning scores in the ICAR (Condon & Revelle, 2014). The correlation was 0.54 [$t(48) = 4.44, p < 0.001, 95\% \text{ CI} = 0.31 \text{ } 0.71$], acceptable for convergent validity purposes (Carlson & Herdman, 2012). Linear regression also showed that correlation did not interact with the order in which participants did the two tasks ($p = 0.716$). The performance standard deviations of the ICAR matrix reasoning items in our sample and in Condon & Revelle (2014) were 0.25 and 0.5, respectively, warranting a correction for age restriction (Condon & Revelle, 2014). The range corrected correlation was 0.79, thus again within acceptable range. To assess the extent of divergent validity, we further compared the correlation described above to the correlations between the ARTOL and the remaining tasks of the ICAR (Table 2), namely, letter-number series completion, verbal reasoning and 3D rotations. The highest correlation with ARTOL performance was indeed with the ICAR matrix reasoning (Table 2). However, comparing the correlations using the Fisher r - z transform revealed that the ARTOL-ICAR matrices correlation did not significantly differ from the others (all $p_s > 0.24$). Taken together, this evidence suggests an acceptable degree of convergent validity between the ARTOL and ICAR-matrix reasoning, with correlation sizes that are similar to those observed between the ICAR and other cognitive ability tests, such as the Shipley 2 (Condon & Revelle, 2014). We observe no evidence of divergent validity from other IQ-related measures of the ICAR, suggesting that the ARTOL may tap into a broad cognitive functioning construct.” (p. 16)

*Table 2. Correlations between ARTOL and tasks from the International Cognitive Ability Resource (“MR” = matrix reasoning, “R3D” = 3D rotations, “LN” = letter and number series completion, “VR” = verbal reasoning). *** $p < 0.001$, ** $p < 0.01$, * $p < 0.05$, ° $p < 0.1$. Bonferroni corrected.*

	ARTOL	MR	R3D	LN
ARTOL				
MR	0.54***			
R3D	0.43*	0.29		
LN	0.39°	0.46**	0.35	
VR	0.45**	0.52**	0.38°	0.37°

2) *I am not convinced that the ART task shows good sensitivity to different age-related differences in reasoning. The results regarding accuracy showed that age effects were driven primarily by the difference between adults and younger adolescents, whereas the task is not really sensitive to age differences between mid and older adolescents. In addition, the measures incorporating response time information (which I consider a critical advantage of the ART) such as the inverse efficiency did not show stable differences, apart from more subtle effects when age was included as a continuous variable. In the absence of an established reasoning measure to evaluate these results against, I find this claim unjustified.*

We clarified our findings in this regard. We no longer state that our task is adequate to identify cross-sectional age differences in reasoning abilities with regards to response times. With regards to accuracy, we have now clarified that we find categorical age differences between mid- and older adolescents only. Importantly, however, we find a robust age effect when analyzing age as a continuous variable. We conclude now that our task is sensitive to age differences in reasoning accuracy (p. 2, 27).

3) *The separation in different age groups was not very clear - whereas the adolescent groups cover a more or less similar age range of 2 - 3 years, the adult includes participants between 18 - 33 years. Given that reasoning development may not be completed by age 18, and most cognitive abilities start declining before age 33 years, it is difficult to use this group as a reference. Looking at the figures, it seems participants in the adult group were not sampled homogeneously across the age range, with only few participants between 18-20 years and > 30 years.*

Many studies have shown that cognitive functions develop relatively quickly during adolescence and then taper off during adulthood. For this reason, it is typical in the developmental literature to sample narrower age ranges during adolescence. The reviewer is right, however, that in particular age brackets sampling was sparse. We now explicitly acknowledge this in the limitations section (p. 26, also copied above in response to point 9 of reviewer 1) so that future study may address this and gradually fill in the gaps of the age continuum. We also agree that the use of age groups is inherently arbitrary (albeit useful). We therefore include a continuous analysis of age for all outcome measures to allow the reader to assess the robustness of our results.

References

- Bolker, B. M., Brooks, M. E., Clark, C. J., Geange, S. W., Poulsen, J. R., Stevens, M. H. H., & White, J.-S. S. (2009). Generalized linear mixed models: a practical guide for ecology and evolution. *Trends in Ecology & Evolution*, *24*(3), 127–135. <http://doi.org/10.1016/J.TREE.2008.10.008>
- Carlson, K. D., & Herdman, A. O. (2012). Understanding the Impact of Convergent Validity on Research Results. *Organizational Research Methods*, *15*(1), 17–32. <http://doi.org/10.1177/1094428110392383>

- Eisinga, R., Te Grotenhuis, M., & Pelzer, B. (2013). The reliability of a two-item scale: Pearson, Cronbach, or Spearman-Brown?. *International journal of public health*, 58(4), 637-642.
- Everitt, B. S., & Skrondal, A. (2002). *The Cambridge dictionary of statistics*. Cambridge.
- Kim, J., & Oshima, T. C. (2013). Effect of Multiple Testing Adjustment in Differential Item Functioning Detection. *Educational and Psychological Measurement*, 73(3), 458-470. <http://doi.org/10.1177/0013164412467033>
- Magis, D., Béland, S., Tuerlinckx, F., & De Boeck, P. (2010). A general framework and an R package for the detection of dichotomous differential item functioning. *Behavior Research Methods*, 42(3), 847-862. <http://doi.org/10.3758/BRM.42.3.847>
- Swaminathan, H., & Rogers, H. J. (1990). Detecting Differential Item Functioning Using Logistic Regression Procedures. *Journal of Educational Measurement*, 27(4), 361-370. <http://doi.org/10.1111/j.1745-3984.1990.tb00754.x>
- Zuur, A. F. (2009). *Mixed effects models and extensions in ecology with R*. Springer.

Appendix C

Most reviewer comments were responded to in this revision. Although some revisions have improved the manuscript, others have not led to the level of standardization, reliability and validity that would be expected for a named measure of fluid intelligence.

Comments

1. Task presentation, p. 10. It is not clear how the items were presented and scored. If no response in 30 seconds, it was scored 0? Also, did subjects have only 8 minutes to complete 80 items? That seems extremely fast. Also, how are item scores and person scores computed when the items are not completed? It seems that these scores are based only on completed items.
2. Given #1, perhaps the most significant problem that remains in this study is the varying completion rates of the items within a set. This leads to two problems. First, accuracy percentage for examinees are based on different numbers of items. Thus, their scores are not comparable. Second, item accuracy scores are based on different examinees (i.e., depending on who reaches the item). Probably the examinees who reach the most items have the greatest ability. Hence, items cannot be compared for accuracy. Further, other statistics (e.g., biserial correlations) are also not comparable between items.
3. Terminology--The term "item difficulty" (p. 2) in the psychometric literature refers to accuracy. Use another term to describe the number of dimensions changing. Also, "Trials" are items? P. 11. If so, label them as such. Puzzle sets in this study are essentially test forms? If so, they should be labeled as such. P. 9
4. Also, p. 17, what is the correlation of "item difficulty" with accuracy and RT? But...if based on variable samples (see #1), this is not feasible.
5. Actually, convergent validity is disappointing.
 - a. First, N is just not sufficient. Although power for significance is OK with 50 subjects, the confidence intervals are too large for the correlation coefficient. Typically, the bare minimum N is 100 for correlations.
 - b. Second, for separate tests of different aptitudes, the mean correlation is usually .70 (often noted as "positive manifold"). One expects higher correlations between tests measuring the same aptitude. $R = .54$ is rather low. The correction for restriction of range involves assumptions that cannot be tested, so the r of .79 cannot be regarded too strongly. Further, of course, the correlation of .45 with verbal reasoning is close the matrix reasoning value. But, again, sample size is just not sufficient.
 - c. Third, while the ICAR is freely-available, it is not a test with extensive validation for measuring fluid intelligence, as is the Raven's. In fact, the main correlates of ICAR are self-reported (!) test scores on other tests (e.g., ASVAB or SAT). Very controversial as self-reports are often not accurate. And also, the other tests are primarily crystallized intelligence measures, not fluid intelligence.
6. P. 12 What are parallel forms defined as here? I thought that "puzzle sets" were designed to be parallel forms, as described under the design section. Thus parallel form reliability only concerns puzzle sets as a predictor.

7. P. 14 How are missing item responses (trials) handled in KR-20? (i.e., discarded due to fast RT). Items apparently have very different N's. And subject total scores will be based on different numbers of items
8. P. 14—Given that the biserial correlations are comparable between items (see comments above), the mean biserial correlation of .31 is fine for an achievement test (heterogeneous content) but not for an aptitude test. I would expect the mean to be .50 or higher. Also, why is the mean p-value .54 when accuracy overall was 69.15%?

Overall, what has been achieved in this study is the thoughtful development of a set of matrix reasoning items. The design of the items is quite rigorous and laudable. However, the psychometric developments are just not there for this to be a named test. Further, the name of the test ARTOL (Abstract Reasoning Task of London) would appear to be an alternate form of a copyrighted test ART (Abstract Reasoning Test), which it clearly is not. A distinctive name for the item bank in the study (not for example APMOL---Advanced Progressive Matrices of London....the Raven's publishers (in London!) would be unhappy) is needed. Matrix Reasoning item bank?

Appendix D

Most reviewer comments were responded to in this revision. Although some revisions have improved the manuscript, others have not led to the level of standardization, reliability and validity that would be expected for a named measure of fluid intelligence.

We thank the reviewer for their thoughtful comments.

Overall, we acknowledge that some psychometric properties of our items are not yet fully developed and we clearly state that we do not intend our task to be used as a measure of fluid intelligence until further psychometric validation has been completed. Additionally, in the new version of our manuscript, we explicitly focus on our items, rather than the task, and have accordingly changed the title of the manuscript to the Matrix Reasoning Item Bank (“MaRs-IB”) (see point 8). We now also highlight more clearly that the MaRs-IB was not implemented with psychometric analyses in mind. Thus the reported psychometric findings are preliminary and further psychometric validation is recommended.

Below are paragraphs of the manuscript that we have now included to make this clearer. These include sections of the abstract and conclusion:

“Working towards an open science framework, we provide 80 novel, open-access abstract reasoning items, an online implementation, and item-level data from 659 participants aged between 11 and 33 years: the Matrix Reasoning Item Bank (“MaRs-IB”) [...]. Further psychometric validation is recommended.” (Abstract, p. 2).

“Our aim here is to introduce a novel, open-access item bank of abstract reasoning items for studies that include adolescents and adults and to provide a preliminary investigation of their psychometric characteristics.” (p. 6)

“Our data stems from a previous study that aimed to address a different research question. The data is therefore not optimized for psychometric validation. In particular, participants completed varying number of items and item-level accuracy data is based on a different number of responses for each item, with completion rates dropping for later items. This may limit their comparability.” (p. 15)

“[...] We therefore recommend a more extensive validation of these items with a representative sample in the future. While we encourage use of the items in assessing non-verbal reasoning in development samples, our item bank should not be considered a normative measure of intelligence until further psychometric testing has been completed.” (p. 29).

“The MaRs-IB is sensitive to age differences in accuracy, which suggests that further psychometric validation will make the item bank a useful resource for researchers interested in abstract reasoning.” (p. 29)

In addition to these clarifications, we have made substantial adjustments to address the reviewer's concerns. For example, to address the sample size issue raised by the reviewer with regard to convergent validity (point 5), we have undertaken a second wave of data collection and provided more analyses assessing convergent validity. We have also run a number of new analyses addressing the reviewer's concerns regarding variable completion rate (see points 2, 4 and 7). These adjustments are described in more detail below.

Comments

1. Task presentation, p. 10. It is not clear how the items were presented and scored. If no response in 30 seconds, it was scored 0? Also, did subjects have only 8 minutes to complete 80 items? That seems extremely fast. Also, how are item scores and person scores computed when the items are not completed? It seems that these scores are based only on completed items.

The reviewer is correct. Our main analyses were based on completed items only. This is now more clearly stated in the manuscript:

"The analyses above were based on completed items. This excludes i) trials in which participants timed out and ii) trials that were not reached." (p. 15).

Regarding the question about time pressure, we have clarified this as follows:

"Participants were not informed of the total number of items and were not required, nor expected, to complete 80 items in 8 minutes. The only time constraint stated was to provide a response within 30 seconds on each trial." (p. 10)

"Timing out occurred in only 2% of items, suggesting that at 30 seconds time limit was feasible." (p. 16)

2. Given #1, perhaps the most significant problem that remains in this study is the varying completion rates of the items within a set. This leads to two problems. First, accuracy percentage for examinees are based on different numbers of items. Thus, their scores are not comparable. Second, item accuracy scores are based on different examinees (i.e., depending on who reaches the item). Probably the examinees who reach the most items have the greatest ability. Hence, items cannot be compared for accuracy. Further, other statistics (e.g., biserial correlations) are also not comparable between items.

The reviewer is correct that accuracy percentage was based on a different number of items. This limitation is now explicitly acknowledged in the manuscript:

"Our data stems from a previous study that aimed to address a different research question. The data is therefore not optimized for psychometric validation. In particular, participants completed varying number of items and item-level accuracy

data is based on a different number of responses for each item, with completion rates dropping for later items. This may limit their comparability.” (p. 15)

To deal with the inherent missingness in our data, we employed mixed models, which, in contrast to using average scores for each participant, were based on trial-level performance and account for differences in the number of trials for each participant (through the participant-level partitioning of the error term).

Furthermore, we re-ran the psychometric tests that were requested by the reviewer (i.e., KR-20, biserial correlations, p-values and differential item functioning) on a subset of items with comparable completion rate. To achieve this, we focused on the 25 items following the first 5 items (since these were deliberately easier). They varied in terms of dimensionality, with dimensionality scores ranging between 1 and 7. To minimize missing values (also see point 7), we here included those trials in which participants had timed out (labeling them as incorrect), as well as those in which response latencies were shorter than 250 ms (only 2 of which were correct).

Doing so solves both of the issues raised by the reviewer, in that for this subset of items: 1) accuracy percentages per participant are no longer based on a different number of items; 2) item-level statistics are no longer based on different participants.

These points are illustrated in the manuscript as follows:

“To address this issue, for the analyses above we used mixed models, which are able to deal with unbalanced data by modeling performance at the trial-, rather than participant-level. Furthermore, to provide preliminary insight into more traditional psychometric approaches to item functioning (i.e. internal consistency, biserial correlations, p-values and differential item functioning), we focused on a subset of data that involved no variability in completion rate.” (p. 14).

“Specifically, we focused on the 25 items following item 5 (since the first 5 items were deliberately easier and involved virtually no variance in performance; see task description). For this analysis we also included trials in which participants had timed out (labeling them as incorrect), as well as those in which response latencies were shorter than 250 ms (only 2 of which were correct). This resulted in a subset of 25 items that were completed by N = 349 participants (at least N = 70 per age group). To assess internal consistency on this subset of items, we computed Kuder-Richardson 20 formula, which resulted in a KR-20 of 0.78 (0.95% CI [0.78, 0.79]). The mean item-total biserial correlation for the same items was 0.32 (SE = 0.02) (biserial correlation of each item is available in Supplementary Table S9), thus in acceptable range, if not for an aptitude test, for a non-verbal reasoning item bank (Everitt & Skrandal, 2002). The mean item p-value of this subset of items (i.e., the proportion of participants that answered correctly to each item) was 0.59 (SE = 0.04) (p-values of each item are available in the Supplementary Tables S9 and S10)[...]. None of these items displayed uniform or non-uniform differential item

functioning, suggesting they are unbiased relative to the age groups studied here and relative to gender. Researchers may want to preferentially use this subset of items for which a more in-depth psychometric analysis is available.” (p. 17)

“In particular, we did not have enough observations to assess the psychometric properties of some of the later items of the task (which few participants reached). While we were able to provide evidence for acceptable internal consistency in a subset of earlier items that involved no variability in completion rate, further psychometric validation of the MaRs-IB is recommended [...]. (p. 29)

In summary, while the reviewer is correct that our analyses involved variation in completion rate, this did not appear to substantially alter the results. Nonetheless, as mentioned above, this feature is now clearly acknowledged as a limitation of the current implementation of our items.

3. Terminology--The term “item difficulty” (p. 2) in the psychometric literature refers to accuracy. Use another term to describe the number of dimensions changing. Also, “Trials” are items? P. 11. If so, label them as such. Puzzle sets in this study are essentially test forms? If so, they should be labeled as such. P. 9

We thank the reviewer for the clarification. We have followed the suggestion. Specifically, we have replaced all occurrences of the term “difficulty” with “dimensionality” with the term “trial” with “item” and “puzzle sets” with “test forms”.

4. Also, p. 17, what is the correlation of “item difficulty” with accuracy and RT? But....if based on variable samples (see #1), this is not feasible.

We presented results of two item-level linear regressions (using mixed models) predicting accuracy and response times based on item dimensionality. We reported the results of the omnibus tests and the regression coefficients. We also have re-run this analysis on items without variable completion rates (i.e., the same subset of items as in point 2) and the results were qualitatively unaltered: item dimensionality continued to negatively predict accuracy ($\chi^2(1) = 408.7$, $b = -0.26$, $SE = 0.01$, $p < 0.001$), and positively predict response time ($\chi^2(1) = 477.03$, $b = 0.09$, $SE = 0.01$, $p < 0.001$).

5. Actually, convergent validity is disappointing.
 - a. First, N is just not sufficient. Although power for significance is OK with 50 subjects, the confidence intervals are too large for the correlation coefficient. Typically, the bare minimum N is 100 for correlations.
 - b. Second, for separate tests of different aptitudes, the mean correlation is usually .70 (often noted as “positive manifold”). One expects higher

correlations between tests measuring the same aptitude. $R = .54$ is rather low. The correction for restriction of range involves assumptions that cannot be tested, so the r of $.79$ cannot be regarded too strongly. Further, of course, the correlation of $.45$ with verbal reasoning is close the matrix reasoning value. But, again, sample size is just not sufficient.

- c. Third, while the ICAR is freely-available, it is not a test with extensive validation for measuring fluid intelligence, as is the Raven's. In fact, the main correlates of ICAR are self-reported (!) test scores on other tests (e.g., ASVAB or SAT). Very controversial as self-reports are often not accurate. And also, the other tests are primarily crystallized intelligence measures, not fluid intelligence.

To address these issues, we increased the sample size to $N = 100$, as recommended by the reviewer (point a). The correlation between the MaRs-IB and the ICAR matrix-reasoning task is now 0.61 . This further provides evidence of divergent validity from a subset of other ICAR tasks (see below) (point b). We have also further justified our use of range restriction whilst acknowledging that some of its assumptions cannot be fully tested, and suggest that the range corrected correlation should be interpreted with caution.

As to the last point (c), we agree that the ICAR is not as well validated as more traditional tasks, especially with regards to external validity (e.g., SATs were indeed self-reported). However, the ICAR has been tested on a large sample of individuals ($> 90,000$) and has at least been indirectly related (through its relation to Shipley-2) to more traditional tests such as Wechsler Adult Intelligence Scale. This is now stated in the manuscript.

Finally, and perhaps most importantly, we are now more explicit in acknowledging that the MaRs-IB is primarily intended as an item bank rather than a fully developed and psychometrically validated task (see our introductory response and point 8 below).

The previously reported passages on convergent validity have been moved to the supplementary material (for transparency on multiple rounds of data collection) and have been replaced with the following:

“A power analysis suggested that 38 participants are sufficient to detect a correlation of 0.5 at 90% power. For this study, we thus initially recruited 50 participants, with a further 50 participants tested upon reviewer request (total $N = 100$, 73 females, 27 males, mean age $=23.95$, $SE = 0.35$, age range 19-35). The results of the initial analysis ($N = 50$) are available in the supplementary material (see supplementary information SI1).” (p. 7).

“The ICAR is relatively novel. While more traditional tasks are more thoroughly validated, the ICAR is computer-based and has been tested on a large sample of

individuals (> 90.000). The ICAR has also at least been indirectly related (through its relation to Shipley-2) to more traditional tests such as the Wechsler Adult Intelligence Scale.” (p. 11)

“Although the assumptions for correcting for range restrictions could not be fully assessed, our sample was tested in a highly selective university. We thus followed Condon and Revelle (2014) and assessed the possible presence of range restriction by comparing the standard deviations of the matrix reasoning items in our sample and theirs. We obtained the latter by combining the standard deviations for the ICAR matrix reasoning items (listed in Table 2 of p. 55 of study by Condon & Revelle)” (p. 15-16).

“To assess convergent validity we inspected the product-moment correlation between MaRs-IB performance (at the aggregate level) and matrix reasoning scores in the ICAR (Condon & Revelle, 2014). The correlation was 0.61 [$t(98) = 7.60, p < 0.001, 95\% \text{ CI} = 0.47 \text{ } 0.72$] and linear regression showed that it did not interact with the order in which participants did the two tasks ($p = 0.58$). Performance standard deviations of the ICAR matrix reasoning items in our sample and in Condon & Revelle (2014) were respectively 0.27 and 0.5, possibly warranting correction for range restriction (Condon & Revelle, 2014). This resulted in a range corrected correlation of 0.81, to be interpreted with caution, given that the assumptions for range restriction correction could not be fully tested. To assess the extent of divergent validity, we further compared the correlation described above to the correlations between the MaRs-IB and the remaining tasks of the ICAR (Table 2), namely, letter-number series completion, verbal reasoning and 3D rotations. The highest correlation with MaRs-IB performance was indeed with the ICAR matrix reasoning (Table 2). Furthermore, Fischer’s $r-z$ transform suggested that this correlation was higher than the correlation between the MaRs-IB and the 3D rotations ($p = 0.03$). However, it was not significantly different from the correlation between the MaRs-IB and letter-number series completion ($p = 0.05$), or between the MaRs-IB and the verbal reasoning task ($p = 0.11$).

Taken together, this evidence suggests that the MaRs-IB correlates strongly with the ICAR’s matrix reasoning task, and more than with another non-verbal reasoning task of the ICAR. However, the uncorrected correlation size falls slightly short of the correlation size that could be expected from the positive manifold of fluid intelligence tests (Condon & Revelle, 2014). We also observe no clear evidence of divergent validity with regards to the ICAR’s verbal reasoning task. We thus recommend further psychometric testing of our items, possibly using more established measures such as Raven’s Matrices (Raven, 2009) and larger samples.” (p. 19-20)

*Table 2. Correlations between performance on MaRs-IB items and tasks from the International Cognitive Ability Resource (“MR” = matrix reasoning, “R3D” = 3D rotations, “LN” = letter and number series completion, “VR” = verbal reasoning). *** $p < 0.001$, ** $p < 0.01$, * $p < 0.05$, ° $p < 0.1$. Bonferroni corrected.*

	ARTOL	MR	R3D	LN
ARTOL				
MR	0.61***			
R3D	0.45***	0.50***		
LN	0.44***	0.54***	0.45***	
VR	0.48***	0.47***	0.29*	0.45***

6. P. 12 What are parallel forms defined as here? I thought that “puzzle sets” were designed to be parallel forms, as described under the design section. Thus parallel form reliability only concerns puzzle sets as a predictor.

We have now clarified this point as follows:

“To assess parallel-forms reliability of the MaRs-IB we used exploratory GLMMs specified as described above to investigate the effect of test form (test form 1, 2 or 3) on accuracy and response times. Moreover, given that test forms were created by pseudo-randomly drawing items from one of three different shape sets, and solutions were pseudo-randomly generated according to one of two distractor strategies, we additionally investigated whether such test form constituents might also independently affect accuracy and response times. This information may be useful for researchers interested in shaping novel test forms (e.g., by differently combining shape sets and distractor strategies). As a secondary parallel forms reliability analysis we thus assessed whether distractor type (minimal or paired difference) and shape set (shape set 1, 2 or 3) might also affect accuracy and response times.” (p. 14)

7. P. 14 How are missing item responses (trials) handled in KR-20? (i.e., discarded due to fast RT). Items apparently have very different N’s. And subject total scores will be based on different numbers of items

To avoid variance in completion rate (and thus different N’s) we have now focused the K2-20 on a subset of items and participants such that all items were completed by all participants. Trials with fast RTs were also re-included to avoid missing values (however, all but two such fast RT responses were incorrect). In spite of these differences, effects on the KR-20 appeared negligible (i.e., from 0.71 to 0.74). The section in question has now been replaced with the one described in point 2.

P. 14—Given that the biserial correlations are comparable between items (see comments above), the mean biserial correlation of .31 is fine for an achievement test (heterogeneous content) but not for an aptitude test. I would expect the mean to be .50 or higher. Also, why is the mean p-value .54 when accuracy overall was 69.15%?

As described in point 2, we have re-run bi-serial correlations on a subset of items that were completed by the same participants, yet the difference in the mean biserial correlation was negligible (i.e., from 0.31 to 0.32). Furthermore, we followed the reviewer's advice in describing this result:

"The mean item-total biserial correlation for the same items was 0.32 (SE = 0.02) (biserial correlation of each item is available in Supplementary Table S9), thus in acceptable range, if not for an aptitude test, for a non-verbal reasoning item bank (Everitt & Skrondal, 2002)" (p. 16)

As per the difference between mean accuracy and the p-value, this is now clarified as follows:

"Note that this mean p-value differs slightly the mean accuracy value reported above because it refers to the subset of items, not to the full dataset." (p.16)

8. Overall, what has been achieved in this study is the thoughtful development of a set of matrix reasoning items. The design of the items is quite rigorous and laudable. However, the psychometric developments are just not there for this to be a named test. Further, the name of the test ARTOL (Abstract Reasoning Task of London) would appear to be an alternate form of a copyrighted test ART (Abstract Reasoning Test), which it clearly is not. A distinctive name for the item bank in the study (not for example APMOL--- Advanced Progressive Matrices of London....the Raven's publishers (in London!) would be unhappy) is needed. Matrix Reasoning item bank? [M-RIB, downplay language a bit... item bank instead of task.]

We thank the reviewer for this suggestion, and have changed the title of our manuscript. Furthermore, throughout our manuscript, we now emphasize the items, rather than the test. We also state that psychometric analyses are preliminary and that more psychometric validation is recommended (please also see our introductory response).

Appendix E

The manuscript has been substantially revised. Appropriate caveats were added, the materials were renamed as an item bank and some additional analyses are included. Both the abstract and conclusion are fine, as is. However, I have identified two remaining issues on wording.

1. The following statement is a good caveat, but the term “assessing” indicates greater validity than is available.

“While we encourage use of the items **in assessing** non-verbal reasoning in development samples, our item bank should not be considered a normative measure of intelligence until further psychometric testing has been completed.” (p. 29).”

The following statement is more appropriate....”While we encourage use of the items in measuring non-verbal reasoning.....”.

2. The following statement is misleading in terms of classical test theory criteria. That is, the forms are not parallel in terms of traditional criteria of item matching, rather than randomization.

“To assess **parallel-forms reliability** of the MaRs-IB we used exploratory GLMMs specified as described above to investigate the effect of test form (test form 1, 2 or 3) on accuracy and response times. Moreover, given that test forms were created by pseudo-randomly drawing items from one of three different shapeS...”

The following statement is more appropriate...”To assess the equivalency of the forms of the MaRs-IB...”

Appendix F

We thank the reviewer again for the helpful comments. We have adjusted the remaining wording issues, the details of which are provided below.

The manuscript has been substantially revised. Appropriate caveats were added, the materials were renamed as an item bank and some additional analyses are included. Both the abstract and conclusion are fine, as is. However, I have identified two remaining issues on wording.

1. The following statement is a good caveat, but the term “assessing” indicates greater validity than is available.

“While we encourage use of the items in assessing non-verbal reasoning in development samples, our item bank should not be considered a normative measure of intelligence until further psychometric testing has been completed.” (p. 29).”

The following statement is more appropriate....”While we encourage use of the items in measuring non-verbal reasoning.....”.

We have followed the reviewer’s advice and replaced the word “assessing” with “measuring” (p. 30).

2. The following statement is misleading in terms of classical test theory criteria. That is, the forms are not parallel in terms of traditional criteria of item matching, rather than randomization.

“To assess parallel-forms reliability of the MaRs-IB we used exploratory GLMMs specified as described above to investigate the effect of test form (test form 1, 2 or 3) on accuracy and response times. Moreover, given that test forms were created by pseudo-randomly drawing items from one of three different shapeS...”

The following statement is more appropriate...”To assess the equivalency of the forms of the MaRs-IB...”

We have replaced this and other occurrences of the words “parallel forms reliability” with those suggested by the reviewer (two occurrences on p. 14, two on p. 18).